
# Interrelations among frustration-free models
# via Witten's conjugation

**Jurriaan Wouters[1*], Hosho Katsura[2,3,4†] and Dirk Schuricht[1‡]**

**1** Institute for Theoretical Physics, Center for Extreme Matter and Emergent Phenomena,
Utrecht University, Princetonplein 5, 3584 CE Utrecht, The Netherlands
**2** Department of Physics, Graduate School of Science, The University of Tokyo,
**3** Institute for Physics of Intelligence, The University of Tokyo,
**4** Trans-scale Quantum Science Institute, The University of Tokyo,
7-3-1, Hongo, Bunkyo-ku, Tokyo, 113-0033, Japan

* j.j.wouters@uu.nl, † katsura@phys.s.u-tokyo.ac.jp, ‡ d.schuricht@uu.nl

## Abstract

We apply Witten's conjugation argument [1] to spin chains, where it allows us to derive frustration-free systems and their exact ground states from known results. We particularly focus on $\mathbb{Z}_p$-symmetric models, with the Kitaev and Peschel–Emery line of the axial next-nearest neighbour Ising (ANNNI) chain being the simplest examples. The approach allows us to treat two $\mathbb{Z}_3$-invariant frustration-free parafermion chains, recently derived by Iemini et al. [2] and Mahyaeh and Ardonne [3], respectively, in a unified framework. We derive several other frustration-free models and their exact ground states, including $\mathbb{Z}_4$- and $\mathbb{Z}_6$-symmetric generalisations of the frustration-free ANNNI chain.



# 1  Introduction

Strongly correlated quantum systems are notoriously hard to study. Even when restricted to one spatial dimension the applicability of analytical methods is rather limited. Notable exceptions are provided by systems like the quantum Ising or XY spin chain that can be mapped to effectively non-interacting models [4], thus allowing the determination of the full spectrum by elementary means. A second class of systems is provided by integrable models [5]. They also allow the determination of the full energy spectrum, although more sophisticated methods like the algebraic Bethe ansatz [6,7] have to be employed and simple results in a closed form are usually not available. A third class of systems is given by the so-called frustration-free models [8]. These are distinguished by the fact that the ground-state manifold can be given in an exact, closed form. In this paper we will discuss such frustration-free models and present an overarching framework connecting many of them.

One of the first frustration-free models was described by Peschel and Emery [9]. They realised that for a constrained set of couplings the ground state of the axial next-nearest neighbour Ising (ANNNI) [10] model takes the simple form of a product state, thus facilitating the straightforward calculation of correlation functions. Along this Peschel–Emery line the model can be viewed as a deformation of the trivial ferromagnetic Ising model. Several generalisations to other two-dimensional models including the three-state Potts model were discovered in the following [11–14]. Recently, frustration-free models of this type have been investigated in the context of Majorana zero modes [15–17] by employing the original results of Peschel and Emery.

Another famous example of a frustration-free model is the Affleck–Kennedy–Lieb–Tasaki (AKLT) chain [8,18,19], which was originally devised in the context of the Haldane conjecture for integer spin chains [20–22]. The idea to construct a parent Hamiltonian was subsequently used to construct further frustration-free models like the $q$-deformed AKLT model [23,

24], valence bond solids with general Lie group symmetries [25–30], or supersymmetric systems [31,32]. As the ground state of the AKLT model can be written as a compact matrix product state it has served as the starting point for the development of the general theory of matrix product and tensor network states [33–38] and their application in numerical simulations [39,40] as well as the classification of quantum phases and their symmetry protections [41–45].

Our investigation was motivated in particular by two recent works by Iemini et al. [2] and Mahyaeh and Ardonne [3]. They constructed two different, frustration-free $\mathbb{Z}_3$-clock models. The motivation for these studies was given by their relation to parafermions, thus naturally generalising Majorana zero modes to $\mathbb{Z}_3$-symmetric systems [46]. Like in the case of the Peschel–Emery line discussed above, both models can be viewed as deformations of a simple classical system, in this case the three-state zero-bias Potts chain. One of our main results is to reformulate both models in a unified framework, thus treating them on an equal footing and clarifying their relation (illustrated in Figure 3).

This will be achieved by applying Witten's conjugation argument [1,47], originally introduced for supersymmetric systems, to spin chains. Starting from a simple model with known ground-state manifold, we derive interacting deformations as well as their exact ground states. The explicit construction then allows the calculation of correlation functions and, in some cases, the proof of the existence of an energy gap. We will apply this line of argument to $\mathbb{Z}_p$-symmetric systems, with the two specific $\mathbb{Z}_3$-symmetric models mentioned above analysed in detail. Furthermore, we construct several new frustration-free models, including generalisations of the Peschel–Emery line to $\mathbb{Z}_4$- and $\mathbb{Z}_6$-symmetric systems.

In this context we note that a method very similar to the Witten conjugation has been applied in the field of matrix product states to construct frustration-free models from the respective parent Hamiltonians [42,43,48,49]. The framework of matrix product (or generalised valence bond solid) states also allows for the calculation of correlation functions, and provides the starting point to prove the existence of an energy gap for the corresponding parent Hamiltonians. These proofs are based either on the martingale method [50] or finite-size criteria [33,51–53]. The latter link the energy gap of a finite-size system to a lower bound on the energy gap in the thermodynamic limit. The first work following such an approach was done by Knabe [54], who used exact diagonalisation on finite-size systems to obtain a lower bound for the energy gap in the spin-1 AKLT model. We will use this approach to obtain bounds for the energy gaps of several models considered in our manuscript. We note that our proofs can in principle be extended by using more advanced methods [50,52], however, the obtained bounds are physically less practical as we discuss for instance for the model in Section 6.5. In order to keep our discussion less abstract, we thus take a more explicit approach not relying on matrix product states in the following but note that many of the results we present below can be rephrased in such terms.

This article is organised as follows: In the next section we discuss Witten's conjugation argument and tailor it to frustration-free spin chains. Section 3 recalls some known families of frustration-free models that are rederived using the deformation approach. In Section 4 we introduce the necessary notations to discuss $\mathbb{Z}_p$-symmetric clock models. In Sections 5 and 6 we analyse two types of deformations, in particular covering the models introduced in References [2,3] in the special case $p = 3$. In addition, we consider several frustration-free $\mathbb{Z}_p$-models. While Witten's conjugation argument applied here ensures the form of the ground state, it does not guarantee the existence of an energy gap. Therefore, in the appendix we apply Knabe's method [54] to obtain lower bounds for the energy gap for some of the considered models.

## 2 Conjugation argument

Originally [1] Witten introduced his conjugation argument in the context of supersymmetric quantum mechanical models. More specifically he discussed, given a supersymmetric Hamiltonian $H$, how to construct an inequivalent Hamiltonian $\tilde{H}$ with the same number of zero-energy states. In this section we recall this argument, already tailoring the notation to the spin-chain systems we will discuss in the following sections. For completeness we recall Witten's original argument in Appendix A.

We consider a lattice with a finite-dimensional Hilbert space for each of the lattice sites. More specifically, in this work we restrict ourselves to one-dimensional chains with open boundary conditions, and assume the local Hamiltonian to act non-trivially neighbouring sites only. We note, however, that the argument presented here is applicable more generally, for example, to periodic boundary conditions, higher-dimensional lattices or longer-ranged models. Coming back to our setup, we consider a Hamiltonian of the form

$$H = \sum_{j=1}^{N-1} H_{j,j+1} = \sum_{j=1}^{N-1} L^{\dagger}_{j,j+1} L_{j,j+1}, \tag{1}$$

where each term[1] $H_{j,j+1} = L^{\dagger}_{j,j+1} L_{j,j+1}$ acts non-trivially on the neighbouring lattice sites $j$ and $j+1$ only, and is positive semi-definite, $\langle \Psi | H_{j,j+1} | \Psi \rangle \geq 0$ for all $| \Psi \rangle$. Consequently the ground-state manifold $G$ is spanned by $| \Psi_1 \rangle, \ldots, | \Psi_n \rangle$, $1 \leq n$, with $L_{j,j+1} | \Psi_i \rangle = 0$ for all $j$; in other words, $G$ is the intersection of the kernels of the operators $L_{j,j+1}$, $G = \bigcap_j \ker(L_{j,j+1})$.

The representation (1) now allows us to say something about the ground states of a deformed/conjugated Hamiltonian. Consider an invertible operator $M_j$ that acts non-trivially on the local Hilbert space of lattice site $j$ only, with which we define an invertible operator acting non-trivially on the whole chain via $M = \prod_j M_j$. Using this operator we can write down the conjugated operators as

$$\tilde{L}_{j,j+1} = M L_{j,j+1} M^{-1} = M_j M_{j+1} L_{j,j+1} M^{-1}_{j+1} M^{-1}_j, \tag{2}$$

where we used $[L_{j,j+1}, M_k] = 0$ for $k \neq j, j+1$. Now the deformed/conjugated local Hamiltonian is given by

$$\tilde{H} = \sum_{j=1}^{N-1} \tilde{H}_{j,j+1} = \sum_{j=1}^{N-1} \tilde{L}^{\dagger}_{j,j+1} C_{j,j+1} \tilde{L}_{j,j+1}, \tag{3}$$

where we have introduced the hermitian operator $C_{j,j+1}$ as additional degrees of freedom in the construction. The operator $C_{j,j+1} = K^{\dagger}_{j,j+1} K_{j,j+1}$ is assumed to be positive definite, $\langle \Psi | C_{j,j+1} | \Psi \rangle > 0$ for all $| \Psi \rangle$, and thus invertible. The product form of $M$ and the locality of $C_{j,j+1}$ and $L_{j,j+1}$ ensure that the resulting Hamiltonian is still local. Note that in general there is no unique annihilation operator $L_{j,j+1}$. Later in this section we will discuss the interplay between the freedom of $C_{j,j+1}$, $L_{j,j+1}$ and $M_j$.

In this setting we can now prove the following theorem (see Reference [1] and Appendix A for the original supersymmetric case):

**Theorem 1.** *The ground-state manifold $\tilde{G}$ of the conjugated Hamiltonian $\tilde{H}$ is given by $\tilde{G} = \mathrm{span}\{M | \Psi_1 \rangle, \ldots, M | \Psi_n \rangle\}$, thus the ground-state degeneracies of $H$ and $\tilde{H}$ are identical.*

We note that the states $M | \Psi_i \rangle$ do not form an orthonormal basis, but since $M$ is invertible the states $\{M | \Psi_1 \rangle, \ldots, M | \Psi_n \rangle\}$ are linearly independent.

---

[1] We use capital letters to denote operators acting on the Hilbert space of the full chain, with subindices indicating on which lattice sites they act non-trivially.

*Proof.* First we show that since $C_{j,j+1}$ is positive definite we have $\ker(\tilde{H}) = \bigcap_j \ker(\tilde{L}_{j,j+1})$. The proof is simple: Note that a priori $\bigcap_j \ker(\tilde{L}_{j,j+1}) \subseteq \ker(\tilde{H})$. Now suppose $|\Psi\rangle \in \ker(\tilde{H})$, ie, $\tilde{H}|\Psi\rangle = 0$, then

$$\langle\Psi|\tilde{H}|\Psi\rangle = \sum_j \langle\Psi|\tilde{L}_{j,j+1}^\dagger C_{j,j+1}\tilde{L}_{j,j+1}|\Psi\rangle = \sum_j \left\|K_{j,j+1}\tilde{L}_{j,j+1}|\Psi\rangle\right\|^2 = 0. \tag{4}$$

This implies $K_{j,j+1}\tilde{L}_{j,j+1}|\Psi\rangle = 0$ for all $j$, and consequently $K_{j,j+1}^\dagger K_{j,j+1}\tilde{L}_{j,j+1}|\Psi\rangle = 0$. Since $C_{j,j+1} = K_{j,j+1}^\dagger K_{j,j+1}$ is invertible we deduce $\tilde{L}_{j,j+1}|\Psi\rangle = 0$ for all $j$. Thus we have shown that $|\Psi\rangle \in \bigcap_j \ker(\tilde{L}_{j,j+1})$, which implies $\ker(\tilde{H}) \subseteq \bigcap_j \ker(\tilde{L}_{j,j+1})$.

Second we have to show $\bigcap_j \ker(\tilde{L}_{j,j+1}) = \tilde{G}$. Note that for all $1 \leq j \leq N-1$ and $1 \leq i \leq n$ we have

$$\tilde{L}_{j,j+1}M|\Psi_i\rangle = ML_{j,j+1}|\Psi_i\rangle = 0, \tag{5}$$

yielding $\tilde{G} \subseteq \bigcap_j \ker(\tilde{L}_{j,j+1})$. Conversely, suppose $|\tilde{\Psi}\rangle \in \bigcap_j \ker(\tilde{L}_{j,j+1})$, then for all $1 \leq j \leq N-1$ we find

$$\tilde{L}_{j,j+1}|\tilde{\Psi}\rangle = ML_{j,j+1}M^{-1}|\tilde{\Psi}\rangle = 0, \tag{6}$$

from which we conclude that $M^{-1}|\tilde{\Psi}\rangle \in \bigcap_j \ker(L_{j,j+1})$. Consequently we can expand the state as $M^{-1}|\tilde{\Psi}\rangle = \sum_i a_i|\Psi_i\rangle$ with suitable $a_i \in \mathbb{C}$, resulting in

$$|\tilde{\Psi}\rangle = \sum_{i=1}^n a_i M|\Psi_i\rangle \in \tilde{G}. \tag{7}$$

Therefore $\bigcap_j \ker(\tilde{L}_{j,j+1}) \subseteq \tilde{G}$, which together with the above implies $\bigcap_j \ker(\tilde{L}_{j,j+1}) = \tilde{G}$.

Finally, we note that since $\tilde{H}$ is positive semi-definite, its ground-state manifold is given by its kernel (provided it is non-zero), thus resulting in $\tilde{G} = \ker(\tilde{H})$ as had to be shown. $\square$

We stress that the theorem above provides a direct way to determine the ground-state degeneracy of the deformed Hamiltonian. On the other hand, the theorem does not make any statement about the energy gap above the ground-state manifold or the excited states of the model. Thus, in Appendix B we will discuss a separate approach to prove the existence of a finite energy gap for some specific models.

Before applying the theorem to the construction of spin chain models, let us discuss the degree of freedom in the choices for $L_{j,j+1}$, $C_{j,j+1}$ and $M_j$. First, assuming a local Hilbert space of dimension $p$, we have the freedom to perform a local basis transformation[2] $v_j$, with $v_j \in \mathrm{U}(p)$, at each lattice site $j$. Under this the operators $M_j$ transform as

$$M_j \to V_j M_j V_j^\dagger, \quad V_j = 1 \otimes \ldots \otimes 1 \otimes v_j \otimes 1 \otimes \ldots \otimes 1, \tag{8}$$

and accordingly

$$L_{j,j+1} \to V_j V_{j+1} L_{j,j+1} V_j^\dagger V_{j+1}^\dagger, \quad C_{j,j+1} \to V_j V_{j+1} C_{j,j+1} V_j^\dagger V_{j+1}^\dagger. \tag{9}$$

Using this we can always choose a suitable basis in the local Hilbert spaces to simplify $M_j$. Second, recalling that the deformed local Hamiltonian is given by

$$\tilde{H}_{j,j+1} = (M^\dagger)^{-1} L_{j,j+1}^\dagger M^\dagger C_{j,j+1} M L_{j,j+1} M^{-1}, \tag{10}$$

we can also perform a transformation on the bonds between lattice sites $j$ and $j+1$ with $u_{j,j+1} \in \mathrm{U}(p^2)$. Specifically setting

$$L_{j,j+1} \to U_{j,j+1} L_{j,j+1}, \quad C_{j,j+1} \to (M^\dagger)^{-1} U_{j,j+1} M^\dagger C_{j,j+1} M U_{j,j+1}^\dagger M^{-1}, \tag{11}$$

---

[2]We use small letters to denote operators acting on the Hilbert space of one or two lattice sites.

where

$$U_{j,j+1} = 1 \otimes \ldots \otimes 1 \otimes u_{j,j+1} \otimes 1 \otimes \ldots \otimes 1, \tag{12}$$

we see that the local Hamiltonian remains invariant. In the examples in the following sections we will use these freedoms to simplify $L_{j,j+1}$.

# 3 Frustration-free models revisited

In this section we will revisit several known frustration-free models within the framework of Witten's conjugation. We first consider two spin-1/2 models: the XY model [55–57] with transverse magnetic field and the ANNNI model [9, 15]. Then we review the $q$-deformed XXZ chain [58–60], and finally we consider the $q$-deformed AKLT model [23, 24].

## 3.1 XY chain in transverse magnetic field

We rederive the frustration-free line for the XY model in a magnetic field. Our starting point is the classical Ising chain (which is equivalent to the Kitaev/Majorana chain [61] in the decoupling limit),

$$H_{j,j+1} = 2 - 2\sigma_j^x \sigma_{j+1}^x, \tag{13}$$

with the exact ground states

$$|\Psi_\pm\rangle = \frac{1}{2^{N/2}} \bigotimes_j \left( |\uparrow\rangle_j \pm |\downarrow\rangle_j \right), \tag{14}$$

where $|\uparrow\rangle_j$ and $|\downarrow\rangle_j$ denote the eigenstates of $\sigma_j^z$ with eigenvalues $\pm 1$. We are looking for models that have a $\mathbb{Z}_2$-symmetry generated by $\prod_j \sigma_j^z$. We choose $M_j$ diagonal, real and positive, thus there is only one independent parameter in $M_j$,

$$M_j = 1 \otimes \ldots \otimes 1 \otimes m_j \otimes 1 \otimes \ldots \otimes 1, \quad m_j = \begin{pmatrix} 1 & \\ & r \end{pmatrix}, \quad 0 < r < \infty. \tag{15}$$

The operator $m_j$ acts at lattice site $j$ only, with the matrix representation given in the basis $\{|\uparrow\rangle_j, |\downarrow\rangle_j\}$. Hence the deformed ground states are

$$|\tilde{\Psi}_\pm\rangle = M |\Psi_\pm\rangle. \tag{16}$$

Note that the states above are not orthogonal. Orthonormal ground states are instead given by

$$|\tilde{\Phi}_\pm\rangle = \frac{1}{N_\pm} \left( M |\Psi_+\rangle \pm M |\Psi_-\rangle \right), \tag{17}$$

with suitable normalisations $N_\pm$. If we take

$$L_{j,j+1} = \sigma_j^x - \sigma_{j+1}^x, \quad C_{j,j+1} = 1, \tag{18}$$

the deformation (3) gives the frustration-free line for the Kitaev chain [61], ie, the Jordan–Wigner transform of the XY chain with magnetic field

$$\tilde{H}_{j,j+1}^{(1)} = -J_x \sigma_j^x \sigma_{j+1}^x - J_y \sigma_j^y \sigma_{j+1}^y + \frac{B^{(1)}}{2}(\sigma_j^z + \sigma_{j+1}^z) + \epsilon, \tag{19}$$

with the parameters

$$J_x = \frac{(r + r^{-1})^2}{2}, \quad J_y = \frac{(r - r^{-1})^2}{2}, \quad B^{(1)} = r^2 - r^{-2}, \quad \epsilon = r^2 + r^{-2}, \tag{20}$$

which correspond to the parameters on the Barouch–McCoy circle [55]. Due to Theorem 1 the model $\tilde{H}^{(1)} = \sum_j \tilde{H}^{(1)}_{j,j+1}$ possesses a two-fold degenerate ground state. In Section 5.3 we will discuss the $\mathbb{Z}_3$-generalisation [2] of this model. Section 5 will be dedicated to generalise the construction to arbitrary $\mathbb{Z}_p$-symmetry.

## 3.2 ANNNI model

For the second example we obtain an interacting parent Hamiltonian of (16) by choosing

$$C_{j,j+1} = \frac{r^2}{2} M_j^{-2} M_{j+1}^{-2}, \tag{21}$$

which acts non-trivially on the neighbouring lattice sites $j$ and $j+1$, with $M_j$ and $L_j$ as in Section 3.1. The resulting deformed local Hamiltonian is the ANNNI model

$$\tilde{H}^{(2)}_{j,j+1} = -\sigma^x_j \sigma^x_{j+1} + J_z \sigma^z_j \sigma^z_{j+1} + \frac{B^{(2)}}{2} \left( \sigma^z_j + \sigma^z_{j+1} \right) + \epsilon, \tag{22}$$

with

$$J_z = \frac{(r - r^{-1})^2}{4}, \quad B^{(2)} = \frac{r^2 - r^{-2}}{2}, \quad \epsilon = \frac{(r + r^{-1})^2}{4}. \tag{23}$$

The frustration-free line rediscovered here is the well-known Peschel–Emery line [9, 15] defined by the relation $B^{(2)} = 2\sqrt{J_z(1 + J_z)}$. The exact two-fold ground-state degeneracy of $\tilde{H}^{(2)} = \sum_j \tilde{H}^{(2)}_{j,j+1}$ is assured by Theorem 1. In Section 6.1.1 we discuss the $\mathbb{Z}_3$-generalisation [3] of this setup, while in Section 6.4 we extend the construction to $\mathbb{Z}_4$-symmetry.

By construction the models (19) and (22) share the same ground states. Thus their combination is also a parent Hamiltonian,

$$\tilde{H}_{j,j+1} = \alpha_1 \tilde{H}^{(1)}_{j,j+1} + \alpha_2 \tilde{H}^{(2)}_{j,j+1}, \tag{24}$$

as long as $\alpha_i \geq 0$. The parameters in the resulting spin model reproduce the frustration-free condition for the XYZ model [56, 57, 62]. Furthermore, the existence of an energy gap above the ground states for (24) has been proven [15]. We also note that the construction above can be extended to inhomogeneous magnetic fields, in particular with an alternating bias [17], or higher-dimensional systems [63, 64].

Finally, we note that the states (16) allow a straightforward calculation of correlation functions. For example, the two-point function of the Ising order parameter is independently of the separation $j - j'$ given by [55]

$$\frac{\langle \tilde{\Psi}_{\pm} | \sigma^x_j \sigma^x_{j'} | \tilde{\Psi}_{\pm} \rangle}{\langle \tilde{\Psi}_{\pm} | \tilde{\Psi}_{\pm} \rangle} = \frac{4}{(r + r^{-1})^2}, \tag{25}$$

which simplifies to unity at the Ising point ($r = 1$) as expected.

## 3.3 $q$-deformed XXZ chain

As a third example we show that the XXX chain and the $q$-deformed XXZ chain are related via Witten's conjugation. We start with the local Hamiltonian of the spin-1/2 XXX Heisenberg chain

$$H_{j,j+1} = 1 - \left( \sigma^x_j \sigma^x_{j+1} + \sigma^y_j \sigma^y_{j+1} + \sigma^z_j \sigma^z_{j+1} \right). \tag{26}$$

We first note that the local Hamiltonian satisfies $(H_{j,j+1})^2 = 4H_{j,j+1}$, which means that the operators $H_{j,j+1}/4$ act as projectors. Thus we can write

$$\frac{1}{4} H_{j,j+1} = |\text{sing}\rangle_{j,j+1} \langle \text{sing}|_{j,j+1}, \tag{27}$$

with[3] $|\text{sing}\rangle_{j,j+1} = (|\uparrow\rangle_j |\downarrow\rangle_{j+1} - |\downarrow\rangle_j |\uparrow\rangle_{j+1})/\sqrt{2}$ denoting the singlet state on the lattice sites $j$ and $j+1$. On all other lattice sites $H_{j,j+1}$ acts trivially. To make the link to the notion introduced above we write

$$\frac{1}{4} H_{j,j+1} = L_{j,j+1}^{\dagger} L_{j,j+1}, \quad L_{j,j+1} = |\uparrow\rangle_j |\downarrow\rangle_{j+1} \langle \text{sing}|_{j,j+1}. \tag{28}$$

Next we consider the generators of $U_q(\text{sl}_2)$ [65]

$$q^{S^z}, \quad S_q^{\pm} = \sum_{j=1}^{N} q^{\sigma_1^z/2} \cdots q^{\sigma_{j-1}^z/2} \sigma_j^{\pm} q^{-\sigma_{j+1}^z/2} \cdots q^{-\sigma_N^z/2}, \tag{29}$$

where we assume $q \in \mathbb{R}$, $q > 0$, and

$$S^z = \frac{1}{2} \sum_{j=1}^{N} \sigma_j^z, \quad \sigma_j^{\pm} = \frac{\sigma_j^x \pm i\sigma_j^y}{2}. \tag{30}$$

These generators satisfy the algebra

$$q^{S^z} S_q^{\pm} q^{-S^z} = q^{\pm 1} S_q^{\pm}, \quad [S_q^+, S_q^-] = \frac{q^{2S^z} - q^{-2S^z}}{q - q^{-1}}, \tag{31}$$

which reduce to the standard relations among the generators of SU(2) in the limit $q \to 1$.

In order to proceed, we next define the operator $M$ via

$$M(q) = q^{-\sigma_1^z/2} \cdots q^{-j\sigma_j^z/2} \cdots q^{-N\sigma_N^z/2}, \tag{32}$$

with the inverse given by

$$M(q)^{-1} = q^{\sigma_1^z/2} \cdots q^{j\sigma_j^z/2} \cdots q^{N\sigma_N^z/2}. \tag{33}$$

With this one gets

$$\tilde{L}_{j,j+1} = M(q) L_{j,j+1} M(q)^{-1} = \sqrt{\frac{1+q^2}{2}} |\uparrow\rangle_j |\downarrow\rangle_{j+1} \langle \text{sing}(q)|_{j,j+1}, \tag{34}$$

where the $q$-deformed singlet state is given by

$$|\text{sing}(q)\rangle_{j,j+1} = \frac{1}{\sqrt{q+q^{-1}}} (q^{-1/2} |\uparrow\rangle_j |\downarrow\rangle_{j+1} - q^{1/2} |\downarrow\rangle_j |\uparrow\rangle_{j+1}). \tag{35}$$

Thus we obtain

$$\tilde{L}_{j,j+1}^{\dagger} \tilde{L}_{j,j+1} = \frac{1+q^2}{2} |\text{sing}(q)\rangle_{j,j+1} \langle \text{sing}(q)|_{j,j+1}, \tag{36}$$

which is manifestly $U_q(\text{sl}_2)$ invariant as it is the projection onto the $q$-deformed singlet state on the bond $(j, j+1)$. A straightforward calculation choosing $C_{j,j+1} = 1$ shows that

$$\frac{1}{4} \tilde{H}_{j,j+1} = \tilde{L}_{j,j+1}^{\dagger} \tilde{L}_{j,j+1} \tag{37}$$

$$= -\frac{q}{4} \left[ \sigma_j^x \sigma_{j+1}^x + \sigma_j^y \sigma_{j+1}^y + \frac{q+q^{-1}}{2} (\sigma_j^z \sigma_{j+1}^z - 1) + \frac{q - q^{-1}}{2} (\sigma_j^z - \sigma_{j+1}^z) \right], \tag{38}$$

---

[3]For the tensor product of states on neighbouring lattice sites we use the short-hand notation $|\uparrow\rangle_j |\downarrow\rangle_{j+1} = |\uparrow\rangle_j \otimes |\downarrow\rangle_{j+1}$ and so on.

which, up to the prefactor $q$, is the local Hamiltonian of the $q$-deformed XXZ chain [58–60,65].

After deriving the Hamiltonian, let us consider the ground states in more detail. The ground states of the Heisenberg chain (26) are given by[4]

$$(S_1^-)^i \ket{\Uparrow}, \quad i = 0, 1, \ldots, N, \quad \text{with} \quad \ket{\Uparrow} = \ket{\uparrow \cdots \uparrow}. \tag{39}$$

Consequently, according to Theorem 1 the ground states of the $q$-deformed model read

$$M(q)(S_1^-)^i \ket{\Uparrow} \propto (\tilde{S}_1^-)^i \ket{\Uparrow}, \quad \text{with} \quad \tilde{S}_1^- = M(q)S_1^- M(q)^{-1} = \sum_{j=1}^{N} q^j \sigma_j^-. \tag{40}$$

However, the $U_q(\mathrm{sl}_2)$ algebra dictates that the ground-state manifold is spanned by

$$(S_q^-)^i \ket{\Uparrow}. \tag{41}$$

By induction we will show that there is a correspondence (up to normalisation) between these sets of states, ie,

$$(S_q^-)^i \ket{\Uparrow} \propto (\tilde{S}_1^-)^i \ket{\Uparrow}. \tag{42}$$

Obviously this relation holds for $i = 0$. Now suppose that (42) is true up to $i - 1$. If we write

$$S_q^\pm = q^{-S^z \pm \frac{1}{2}} \sum_{j=1}^{N} q^{\sigma_1^z} \cdots q^{\sigma_{j-1}^z} \sigma_j^\pm, \tag{43}$$

then

$$(S_q^-)^i \ket{\Uparrow} \propto S_q^- (\tilde{S}_1^-)^{i-1} \ket{\Uparrow} = (i-1)! S_q^- \sum_{j_1 < \cdots < j_{i-1}} q^{j_1 + \cdots + j_{i-1}} \sigma_{j_1}^- \cdots \sigma_{j_{i-1}}^- \ket{\Uparrow} \tag{44}$$

$$= (i-1)! q^{-\frac{N+1}{2}} \frac{q^i - q^{-i}}{q - q^{-1}} \sum_{j_1 < \cdots < j_i} q^{j_1 + \cdots + j_i} \sigma_{j_1}^- \cdots \sigma_{j_i}^- \ket{\Uparrow} \tag{45}$$

$$= \frac{q^{-\frac{N+1}{2}}}{i} \frac{q^i - q^{-i}}{q - q^{-1}} (\tilde{S}_1^-)^i \ket{\Uparrow}, \tag{46}$$

where the precise prefactor is in fact irrelevant for our purpose. This shows that the relation (42) is indeed fulfilled, and thus that the ground states of the $q$-deformed model are given by $M(q)(S_1^-)^i \ket{\Uparrow}$.

### 3.4 $q$-deformed AKLT chain

Arguably one of the most prominent frustration-free models is the AKLT chain [8,18,19]. Even though the ground state of this system is a matrix product state, we will see that we can still employ the tools outlined above to derive its $q$-deformed generalisation [23,24,66].

We start with the original AKLT chain written as

$$H = \sum_{j} H_{j,j+1}, \quad H_{j,j+1} \equiv \sum_{m=-2}^{2} \ket{\psi_m}_{j,j+1} \bra{\psi_m}_{j,j+1}, \tag{47}$$

where $H_{j,j+1}$ is the projector onto the subspace of total spin-2 on the neighbouring lattice sites $j$ and $j + 1$. It can be written in terms of the corresponding eigenstates $\ket{\psi_m}_{j,j+1}$ and acts

---

[4]We note that the subscript refers to the deformation parameter, ie, $S_1^- \equiv S_{q=1}^-$.

trivially on all other lattice sites. The eigenstates are explicitly given by

$$|\psi_2\rangle_{j,j+1} = |+\rangle_j |+\rangle_{j+1}, \quad |\psi_1\rangle_{j,j+1} = \frac{1}{\sqrt{2}}\left(|+\rangle_j |0\rangle_{j+1} + |0\rangle_j |+\rangle_{j+1}\right),$$

$$|\psi_0\rangle_{j,j+1} = \frac{1}{\sqrt{6}}\left(|+\rangle_j |-\rangle_{j+1} + |-\rangle_j |+\rangle_{j+1} + 2|0\rangle_j |0\rangle_{j+1}\right),$$

$$|\psi_{-1}\rangle_{j,j+1} = \frac{1}{\sqrt{2}}\left(|0\rangle_j |-\rangle_{j+1} + |-\rangle_j |0\rangle_{j+1}\right), \quad |\psi_{-2}\rangle_{j,j+1} = |-\rangle_j |-\rangle_{j+1}, \tag{48}$$

with $|\pm\rangle_j$, $|0\rangle_j$ denoting the eigenstates of the spin-1 operator $S_j^z$ at a given lattice site $j$. Note that since $H_{j,j+1}$ is a projector, we can match our convention by simply setting $L_{j,j+1} = H_{j,j+1}$. For the deformation we choose ($q \in \mathbb{R}$, $q > 0$)

$$M(q) = \prod_j M_j(q), \qquad M_j(q) = q^{-2jS_j^z}\left(\frac{q+q^{-1}}{2}\right)^{(S_j^z)^2/2}, \tag{49}$$

and we define $q$-deformed states

$$|\tilde{\psi}_2^q\rangle_{j,j+1} = |+\rangle_j |+\rangle_{j+1}, \quad |\tilde{\psi}_1^q\rangle_{j,j+1} = \frac{1}{\sqrt{1+q^4}}\left(|+\rangle_j |0\rangle_{j+1} + q^2 |0\rangle_j |+\rangle_{j+1}\right),$$

$$|\tilde{\psi}_0^q\rangle_{j,j+1} = \frac{q^{-2}|+\rangle_j |-\rangle_{j+1} + q^2 |-\rangle_j |+\rangle_{j+1} + (q+q^{-1})|0\rangle_j |0\rangle_{j+1}}{\sqrt{q^4 + q^{-4} + (q+q^{-1})^2}},$$

$$|\tilde{\psi}_{-1}^q\rangle_{j,j+1} = \frac{1}{\sqrt{1+q^4}}\left(|0\rangle_j |-\rangle_{j+1} + q^2 |-\rangle_j |0\rangle_{j+1}\right), \quad |\tilde{\psi}_{-2}^q\rangle_{j,j+1} = |-\rangle_j |-\rangle_{j+1}. \tag{50}$$

We can then work out that the conjugated annihilation operator $\tilde{L}_{j,j+1}$ is given by

$$\tilde{L}_{j,j+1} \equiv |\tilde{\psi}_2^{q^{-1}}\rangle_{j,j+1} \langle\tilde{\psi}_2^q|_{j,j+1} + |\tilde{\psi}_{-2}^{q^{-1}}\rangle_{j,j+1} \langle\tilde{\psi}_{-2}^q|_{j,j+1}$$

$$+ a(q)\left(|\tilde{\psi}_1^{q^{-1}}\rangle_{j,j+1} \langle\tilde{\psi}_1^q|_{j,j+1} + |\tilde{\psi}_{-1}^{q^{-1}}\rangle_{j,j+1} \langle\tilde{\psi}_{-1}^q|_{j,j+1}\right) + b(q)|\tilde{\phi}_0\rangle_{j,j+1} \langle\tilde{\psi}_0^q|_{j,j+1}, \tag{51}$$

with the auxiliary state

$$|\tilde{\phi}_0\rangle_{j,j+1} = q^2 |+\rangle_j |-\rangle_{j+1} + q^{-2}|-\rangle_j |+\rangle_{j+1} + \frac{4}{q+q^{-1}}|0\rangle_j |0\rangle_{j+1} \tag{52}$$

and the parameters

$$a(q) = \frac{q^2 + q^{-2}}{2}, \quad b(q) = \frac{(q^2+q^{-2})(q^2+q^{-2}+1)}{6}. \tag{53}$$

Now we choose $C_{j,j+1}$ as

$$C_{j,j+1} = |\tilde{\psi}_2^{q^{-1}}\rangle_{j,j+1} \langle\tilde{\psi}_2^{q^{-1}}|_{j,j+1} + |\tilde{\psi}_{-2}^{q^{-1}}\rangle_{j,j+1} \langle\tilde{\psi}_{-2}^{q^{-1}}|_{j,j+1}$$

$$+ \frac{1}{a(q)^2}\left(|\tilde{\psi}_1^{q^{-1}}\rangle_{j,j+1} \langle\tilde{\psi}_1^{q^{-1}}|_{j,j+1} + |\tilde{\psi}_{-1}^{q^{-1}}\rangle_{j,j+1} \langle\tilde{\psi}_{-1}^{q^{-1}}|_{j,j+1}\right)$$

$$+ \frac{1}{b(q)^2}|\tilde{\phi}_0\rangle_{j,j+1} \langle\tilde{\phi}_0|_{j,j+1}, \tag{54}$$

such that the deformed local Hamiltonian becomes the projector

$$\tilde{H}_{j,j+1} = \tilde{L}_{j,j+1}^\dagger C_{j,j+1} \tilde{L}_{j,j+1} \equiv \sum_{m=-2}^{2} |\tilde{\psi}_m^q\rangle_{j,j+1} \langle\tilde{\psi}_m^q|_{j,j+1}. \tag{55}$$

Hence we obtain the $q$-deformed AKLT model [23, 24, 66].

The above result shows the deformation at the level of the Hamiltonian. Let us also look explicitly at the ground state. The four ground states of the undeformed AKLT chain can be written in the matrix product state representation as

$$
\begin{pmatrix} |\Psi_{\text{AKLT}}^{1,1}\rangle & |\Psi_{\text{AKLT}}^{1,2}\rangle \\ |\Psi_{\text{AKLT}}^{2,1}\rangle & |\Psi_{\text{AKLT}}^{2,2}\rangle \end{pmatrix} = A_1 \cdots A_L \,, \text{ with } A_j = \begin{pmatrix} |0\rangle_j & -\sqrt{2}\,|+\rangle_j \\ \sqrt{2}\,|-\rangle_j & -|0\rangle_j \end{pmatrix}. \tag{56}
$$

According to Theorem 1, the ground state of the $q$-deformed model is generated by the matrix

$$
\tilde{A}_j = \begin{pmatrix} |0\rangle_j & -q^{-2j}\sqrt{q+q^{-1}}\,|+\rangle_j \\ q^{2j}\sqrt{q+q^{-1}}\,|-\rangle_j & -|0\rangle_j \end{pmatrix}. \tag{57}
$$

Generically a matrix product state is defined up to a gauge freedom. If we take

$$
f_{j-1,j} = \begin{pmatrix} q^j & \\ & q^{-(j-1)} \end{pmatrix}, \tag{58}
$$

we can redefine the matrix representation as

$$
\tilde{A}_j^{\text{tr}} = f_{j-1,j}\tilde{A}_j f_{j,j+1}^{-1} = \begin{pmatrix} q^{-1}\,|0\rangle_j & -\sqrt{q+q^{-1}}\,|+\rangle_j \\ \sqrt{q+q^{-1}}\,|-\rangle_j & -q\,|0\rangle_j \end{pmatrix}, \tag{59}
$$

which is identical to the one given in References [24, 66] for the ground state of the $q$-deformed AKLT chain.

Finally we note that a similar derivation to the one presented in this section can be used to relate the AKLT chain (47) to a frustration-free point in the (representation) symmetry protected phase of $S_3$-invariant chains recently studied by O'Brien et al. [67].

## 4 Introduction to $\mathbb{Z}_p$-clock models

The rest of the paper considers $\mathbb{Z}_p$-clock models and frustration-free systems of this type. Therefore, let us first briefly review the $\mathbb{Z}_p$-clock algebra. Consider a local $p$-dimensional Hilbert space and two local operators $\sigma$ and $\tau$ satisfying

$$
\sigma^p = \tau^p = 1\,, \quad \sigma^{p-1} = \sigma^\dagger\,, \quad \tau^{p-1} = \tau^\dagger\,, \quad \sigma\tau = \omega\tau\sigma\,, \tag{60}
$$

where $\omega = \exp(2\pi i/p)$ is the $p$th root of unity. Denoting the eigenstates of $\sigma$ and $\tau$ by $|\sigma, i\rangle$ and $|\tau, i\rangle$ with $i = 0, \ldots, p-1$ respectively, the action of the operators is given by

$$
\sigma\,|\sigma, i\rangle = \omega^i\,|\sigma, i\rangle\,, \quad \tau\,|\sigma, i\rangle = |\sigma, i+1\rangle\,, \tag{61}
$$

$$
\tau\,|\tau, i\rangle = \omega^i\,|\tau, i\rangle\,, \quad \sigma\,|\tau, i\rangle = |\tau, i-1\rangle\,, \tag{62}
$$

where $i \pm 1$ has to be taken modulo $p$. The states $|\sigma, i\rangle$ can be represented in terms of the states $|\tau, i\rangle$ as

$$
|\sigma, i\rangle = \frac{1}{\sqrt{p}}\left(|\tau, 0\rangle + \omega^i\,|\tau, 1\rangle + \cdots + \omega^{(p-1)i}\,|\tau, p-1\rangle\right). \tag{63}
$$

The Potts/clock model is a generalisation of the Ising model. Here we start with the counterpart of the classical Ising chain (13), namely the classical Potts model, whose local Hamiltonian is given by

$$
H_{j,j+1} = 2 - \sigma_j\sigma_{j+1}^\dagger - \sigma_j^\dagger\sigma_{j+1}\,, \tag{64}
$$

where $\sigma_j$ and $\tau_j$ denote the operators $\sigma$ and $\tau$ introduced above, now acting non-trivially on the local Hilbert space of site $j$. The classical Potts model possesses a $p$-fold degenerate ground state

$$|\Psi_i\rangle = \bigotimes_j |\sigma, i\rangle_j \,, \tag{65}$$

with $|\sigma, i\rangle_j$ denoting the eigenstates of $\sigma_j$. Furthermore, the system has an energy gap above the ground states.

Finally, we note that the clock operators $\sigma_j$ and $\tau_j$ have a parafermionic dual by virtue of the Fradkin–Kadanoff transformation [68], which is the generalisation of the Jordan–Wigner transformation to $\mathbb{Z}_p$-symmetry. The resulting parafermions can be regarded as a generalisation of Majorana fermions [46].

We can already discuss the most general form of deformation that we consider in the rest of the paper. So far the only requirement for $M_j$ is the invertibility. In this work we restrict ourselves to models that preserve $\mathbb{Z}_p$-symmetry generated by

$$\omega^P = \prod_j \tau_j \,. \tag{66}$$

Since $M$ has to commute with $\omega^P$, the local operator $m_j$ has to be diagonal in the $\tau$-basis, ie,

$$m_j = \begin{pmatrix} \alpha_0 & & & \\ & \alpha_1 & & \\ & & \ddots & \\ & & & \alpha_{p-1} \end{pmatrix} = \frac{1}{p} \sum_{k,l=0}^{p-1} \alpha_k \omega^{-kl} \tau^l \,, \tag{67}$$

for $\alpha_k \in \mathbb{C}$. Note that we can take out an overall scaling factor, so we end up with $p-1$ independent complex coefficients $\alpha_k/\alpha_0$, $k = 1, \dots, p-1$. For now we will leave it in the most general form. In line with the cyclicity of the algebra, the coefficients $\alpha_k$ are also defined modulo $p$,

$$\alpha_k = \alpha_{k \bmod p} \,, \tag{68}$$

for instance $\alpha_{-k} = \alpha_{p-k}$. Later we will see that in specific examples we get more constraints on the coefficients $\alpha_k$.

Starting with the ground states (65) we obtain the deformed states by acting with the operator $M = \prod_j M_j = \bigotimes_j m_j$, ie,

$$|\tilde{\Psi}_i\rangle = M |\Psi_i\rangle = \bigotimes_j m_j |\sigma, i\rangle_j \,. \tag{69}$$

This form immediately allows us to calculate correlation functions. For example, the two-point function of the order parameter $\sigma$ becomes

$$\left| \frac{\langle \tilde{\Psi}_i | \sigma_j \sigma_{j'}^\dagger | \tilde{\Psi}_i \rangle}{\langle \tilde{\Psi}_i | \tilde{\Psi}_i \rangle} \right| = \frac{|\sum_k \alpha_k^* \alpha_{k+1}|^2}{(\sum_k |\alpha_k|^2)^2} \leq 1 \,, \tag{70}$$

where the upper bound is obtained by virtue of the Schwarz inequality. Other correlation functions can be obtained in a similar way. In the following sections we will derive the parent Hamiltonian for the deformed ground states.

# 5  Frustration-free $\mathbb{Z}_p$-generalisations of the XY chain

In this section we generalise the $\mathbb{Z}_2$-XY chain discussed in Section 3.1 to arbitrary $\mathbb{Z}_p$-symmetry. Specifically we use the term XY in the sense that we take $L_{j,j+1}$ and $C_{j,j+1}$ of the following form

$$L_{j,j+1} = \sigma_j - \sigma_{j+1}, \quad C_{j,j+1} = 1. \tag{71}$$

Furthermore we require the resulting model to possess $\omega^P$-symmetry, which fixes $m_j$ to be given by (67). In the case $p = 3$ we recover a model recently studied by Iemini et al. [2], see Section 5.2.

For the choices (67) and (71) it is straightforward to work out the conjugated Hamiltonian (we set $\alpha_{-1} \equiv \alpha_{p-1}$ to lighten the notation)

$$\tilde{L}_{j,j+1} = \frac{1}{p} \sum_{k,l=0}^{p-1} \frac{\alpha_{k-1}}{\alpha_k} \omega^{-kl} \left( \sigma_j \tau_j^l - \sigma_{j+1} \tau_{j+1}^l \right), \tag{72}$$

where we used [see Equation (60)]

$$M_j \sigma_j M_j^{-1} = \frac{1}{p^2} \sum_{k,k',l,l'} \frac{\alpha_k}{\alpha_{k'}} \omega^{-(k+1)l - k'l'} \sigma_j \tau_j^{l+l'} = \frac{1}{p} \sum_{k,l=0}^{p-1} \frac{\alpha_{k-1}}{\alpha_k} \omega^{-kl} \sigma_j \tau_j^l. \tag{73}$$

With (72) the conjugated local Hamiltonian then becomes

$$\tilde{H}_{j,j+1} = \frac{1}{p^2} \sum_{k,k',l,l'} \frac{\alpha_{k-1}^*}{\alpha_k^*} \frac{\alpha_{k'-1}}{\alpha_{k'}} \omega^{kl - k'l'} \left[ \left( \tau_j^{l'-l} + \tau_{j+1}^{l'-l} \right) - \left( \tau_j^{-l} \sigma_j^\dagger \sigma_{j+1} \tau_{j+1}^{l'} + \text{h.c.} \right) \right]$$

$$= -\left( B_j^\dagger \sigma_j^\dagger \sigma_{j+1} B_{j+1} + \text{h.c.} \right) + \sum_{l=0}^{p-1} \gamma_l \left( \tau_j^l + \tau_{j+1}^l \right), \tag{74}$$

where

$$B_j = \sum_{l=0}^{p-1} \beta_l \tau_j^l, \quad \beta_l = \frac{1}{p} \sum_{k=0}^{p-1} \frac{\alpha_{k-1}}{\alpha_k} \omega^{-kl}, \quad \gamma_l = \frac{1}{p} \sum_{k=0}^{p-1} \left| \frac{\alpha_{k-1}}{\alpha_k} \right|^2 \omega^{-kl}. \tag{75}$$

Admittedly this form is not yet very insightful. Thus in the following sections we will consider specific cases for which the Hamiltonian simplifies.

## 5.1  $\mathbb{Z}_p$-XY model: most general real coefficients

One simplification occurs with the requirement that the coefficients $\beta_l$ and $\gamma_l$ are real. For odd $p$ this implies the following conditions (we set $\alpha_0 = r_0 = 1$ due to the freedom in the overall scaling of $m_j$)

$$\alpha_k = \begin{cases} e^{\mathrm{i}\theta_k} r_k, & k = 1, \ldots, \frac{p-1}{2}, \\ e^{\mathrm{i}\theta_{p-k-1}} \frac{r_{(p-1)/2}^2}{r_{p-k-1}}, & k = \frac{p+1}{2}, \ldots, p-1, \end{cases} \tag{76}$$

for $r_1, \ldots, r_{(p-1)/2} > 0$ and $\theta_1, \ldots, \theta_{(p-1)/2} \in [0, 2\pi)$. Similarly, for $p$ even $\beta_l$ and $\gamma_l$ are real provided

$$\alpha_k = \begin{cases} e^{\mathrm{i}\theta_k} r_k, & k = 1, \ldots, \frac{p}{2} - 1, \\ \pm e^{\mathrm{i}\theta_{p-k-1}} \frac{s}{r_{p-k-1}}, & k = \frac{p}{2}, \ldots, p-1, \end{cases} \tag{77}$$

for $r_1, \ldots, r_{p/2-1}, s > 0$ and $\theta_1, \ldots, \theta_{p/2-1} \in [0, 2\pi)$.

## 5.2   $\mathbb{Z}_p$-XY model: compact form with real coefficients

In order to obtain a compact form for the Hamiltonian (74) the results from the previous section can be further specified. Taking $\alpha_k = r^k$ with $r \in \mathbb{R}\backslash\{0\}$ such that the ratio between consecutive $\alpha_k$ is constant, we obtain for the coefficients

$$\beta_l = \frac{1}{pr}\left(r^p + p\delta_{l,0} - 1\right), \quad \gamma_l = \frac{1}{pr^2}\left(r^{2p} + p\delta_{l,0} - 1\right). \tag{78}$$

Thus the local Hamiltonian simplifies to

$$\tilde{H}_{j,j+1} = \epsilon - \left[\left(1 + b\sum_{l=1}^{p-1}\tau_j^l\right)\sigma_j^\dagger\sigma_{j+1}\left(1 + b\sum_{l=1}^{p-1}\tau_{j+1}^l\right) + \text{h.c.}\right] - \frac{f}{2}\sum_{l=1}^{p-1}\left(\tau_j^l + \tau_{j+1}^l\right), \tag{79}$$

with

$$b = \frac{r^p - 1}{r^p + p - 1}, \quad f = \frac{2p(1 - r^{2p})}{(r^p + p - 1)^2}, \quad \epsilon = \frac{p(r^{2p} + p - 1)}{(r^p + p - 1)^2}, \tag{80}$$

where we have done a multiplicative rescaling to set the coupling of $\sigma_j^\dagger\sigma_{j+1}$ to $-1$. For $p = 2$ the model simplifies to the XY model (19) discussed in Section 3.1.

We note that for odd $p$ the model parameters depend on the sign of $r$, while for even $p$ the coefficients only contain even powers of $r$. The latter suggests that there are two sets of ground states for the same Hamiltonian,

$$|\tilde{\Psi}_i^+\rangle = M(r)|\Psi_i\rangle, \quad |\tilde{\Psi}_i^-\rangle = M(-r)|\Psi_i\rangle. \tag{81}$$

However, from the expansion we recognise

$$|\tilde{\Psi}_i^-\rangle = |\tilde{\Psi}_{i+p/2}^+\rangle, \tag{82}$$

so both $M(r)$ and $M(-r)$ provide the same set of ground states. Moreover, the physical properties do not change under $r \rightarrow 1/r$. We can see this from $m_j = \text{diag}(1, r, \ldots, r^n) \rightarrow \text{diag}(1, r^{-1}, \ldots, r^{-n}) \propto \text{diag}(r^n, r^{n-1}, \ldots, 1)$. The latter is related to the original $m_j$ by a conjugation and cyclic rotation of the basis, hence the physical properties remain invariant.

Finally we note that for $p = 3$ and $r > 0$ we reproduce the model introduced by Iemini et al. [2]. There the authors also derive the positive-definite form (3) by the use of Fock parafermions [69]. Using elementary methods, in Appendix B.2 we show that the model possesses a finite energy gap for $0.5695 \lesssim r \lesssim 1/0.5695 \approx 1.7560$, thus confirming the corresponding numerical results [2]. We note that our proof does not exclude the existence of an energy gap outside this interval, which can be extended by improving our analysis or using alternative methods [33, 50, 52]. We note, however, that special care has to be taken regarding the treatment of the boundary conditions.

## 5.3   $\mathbb{Z}_3$-XY model: real coefficients from complex deformation

Our construction allows us to directly generalise the model discussed above. From Section 5.1 we see that for $p = 3$ there is an additional freedom in the choice of $m_j$ in the form of a complex phase, ie, we can choose

$$m_j = \begin{pmatrix} 1 & & \\ & e^{i\theta}r & \\ & & r^2 \end{pmatrix}, \tag{83}$$

which results in

$$\tilde{H}_{j,j+1} = \epsilon - \left[ \left( 1 + b^+ \tau_j + b^- \tau_j^\dagger \right) \sigma_j^\dagger \sigma_{j+1} \left( 1 + b^- \tau_{j+1} + b^+ \tau_{j+1}^\dagger \right) + \frac{f}{2} \left( \tau_j + \tau_{j+1} \right) + \text{h.c.} \right],$$

(84)

with the parameters

$$f = \frac{6(1-r^6)}{(r^3 + 2\cos\theta)^2}, \quad b^\pm = \frac{r^3 - \cos\theta \pm \sqrt{3}\sin\theta}{r^3 + 2\cos\theta}, \quad \epsilon = \frac{6(r^6 + 2)}{(r^3 + 2\cos\theta)^2}.$$

(85)

For $\theta = 0$ we recover the model studied in Reference [2]. We note that the parameters (85) possess a divergence at $r = \sqrt[3]{-2\cos\theta}$ provided $\theta \in [\frac{\pi}{2}, \frac{3\pi}{2}]$. This divergence is an artefact of fixing the prefactor of the $-\sigma_j^\dagger \sigma_{j+1}$-term to unity, it can be removed by rescaling the Hamiltonian by $(r^3 + 2\cos\theta)^2$.

# 6 Frustration-free $\mathbb{Z}_p$-generalisations of the ANNNI model

In this section we construct $\mathbb{Z}_p$-invariant generalisations of the ANNNI model (see Section 3.2), for which we will use the term[5] axial next-nearest neighbour Potts (ANNNP) model [14]. More specifically we consider $\mathbb{Z}_p$-invariant Hamiltonians where besides the classical Potts term $\sigma_j \sigma_{j+1}^\dagger + \sigma_j^\dagger \sigma_{j+1}$ only terms of the form $\tau_j^l \tau_{j+1}^{l'}$ with $l, l' = 0, \dots, p-1$ appear. In particular, there are no terms containing products of $\sigma$- and $\tau$-operators.

First we will derive some general results following from this simple set of rules. Then we discuss several specific examples. We take $m_j$ to be defined by (67) and $L_{j,j+1} = \sigma_j - \sigma_{j+1}$ as before. Furthermore, generalising (21) we set $C_{j,j+1} = K_j K_{j+1}$, where $K_j$ acts non-trivially at lattice site $j$ with the matrix $k_j$. Now making the ansatz (in the $\tau$-basis)

$$k_j = \text{diag}\left( \frac{\alpha_1}{\alpha_0}, \frac{\alpha_2}{\alpha_1}, \dots, \frac{\alpha_{p-1}}{\alpha_{p-2}}, \frac{\alpha_0}{\alpha_{p-1}} \right),$$

(86)

and recalling that $C_{j,j+1}$ has to be hermitian and positive definite, we deduce that the $\alpha_k$ have to be real and positive (we set $\alpha_0 = 1$). From the form above we also deduce that the following identity holds, $K_j M_j \sigma_j M_j^{-1} = \sigma_j$. Hence we find for the deformed local Hamiltonian

$$\begin{aligned}
\tilde{H}_{j,j+1} &= \tilde{L}_{j,j+1}^\dagger K_j K_{j+1} \tilde{L}_{j,j+1} = \tilde{L}_{j,j+1}^\dagger \left( \sigma_j K_{j+1} - K_j \sigma_{j+1} \right) \\
&= \left( M_j^{-1} \sigma_j^\dagger M_j - M_{j+1}^{-1} \sigma_{j+1}^\dagger M_{j+1} \right) \left( \sigma_j K_{j+1} - K_j \sigma_{j+1} \right) \\
&= -\left( \sigma_j \sigma_{j+1}^\dagger + \sigma_j^\dagger \sigma_{j+1} \right) + \left( M_j^{-1} \sigma_j^\dagger M_j \sigma_j K_{j+1} + K_j M_{j+1}^{-1} \sigma_{j+1}^\dagger M_{j+1} \sigma_{j+1} \right).
\end{aligned}$$

(87)

Here the first two terms represent the classical Potts model. Note that both $M_j$ and $K_j$ are diagonal in the $\tau$-basis and can therefore be expanded in powers of $\tau_j$.

$$M_j^{-1} \sigma_j^\dagger M_j \sigma_j = \sum_l \Delta_l \tau_j^l, \quad K_j = \sum_l \Gamma_l \tau_j^l,$$

(88)

where we introduced the abbreviations

$$\Delta_l = \frac{1}{p} \sum_k \frac{\alpha_{k-1}}{\alpha_k} \omega^{-kl}, \quad \Gamma_l = \frac{1}{p} \sum_k \frac{\alpha_{k+1}}{\alpha_k} \omega^{-kl}.$$

(89)

---

[5]Alternatively, since the models will be written in terms of the clock operators, we could use the term axial next-nearest neighbour clock (ANNNC) model [70].

Hence, the last two terms in (87) only produce contributions of the form $\tau_j^l \tau_{j+1}^{l'}$, as was intended. We will not write down the explicit expansion, since it is tedious and not insightful. Instead, in the next sections we will discuss several explicit examples. Doing so we obtain a general complex $\mathbb{Z}_3$-ANNNP model. Furthermore, we rediscover the known frustration-free line [3, 14] in the $\mathbb{Z}_3$ case, with purely real coefficients. Finally, we discuss a frustration-free line for $p = 2q$ even, of which the original ANNNI model (22) is the simplest representative and $\mathbb{Z}_4$- and $\mathbb{Z}_6$-ANNNP examples are given below.

## 6.1  $\mathbb{Z}_3$-ANNNP model: with complex coefficients

The simplest non-trivial example (besides ANNNI) we can derive with this construction is the $\mathbb{Z}_3$-ANNNP. The most general deformation for $\mathbb{Z}_3$ is

$$m_j = \begin{pmatrix} 1 & & \\ & r & \\ & & s \end{pmatrix}, \tag{90}$$

with the corresponding $C_{j,j+1}$ determined by (86). The deformed Hamiltonian takes the simple form

$$\tilde{H} = -\sum_j \left[ \sigma_j \sigma_{j+1}^\dagger + \frac{f}{2}(\tau_j + \tau_{j+1}) + g_1 \tau_j \tau_{j+1} + g_2 \tau_j \tau_{j+1}^\dagger + \text{h.c.} \right] + \epsilon, \tag{91}$$

which is also the quantum limit of the axial next-nearest neighbour Potts model [14]. Since the operators $\sigma_j$ and $\tau_j$ are not self-adjoint, more terms and coefficients than in the original ANNNI model (22) appear. The similarity with the ANNNI model is exemplified by the following identifications:

| ANNNI model | | $\mathbb{Z}_3$-ANNNP model |
|---|---|---|
| $\sigma_j^x \sigma_{j+1}^x$ | $\rightarrow$ | $\sigma_j \sigma_{j+1}^\dagger$ |
| $\sigma_j^z$ | $\rightarrow$ | $\tau_j$ |
| $\sigma_j^z \sigma_{j+1}^z$ | $\rightarrow$ | $\tau_j \tau_{j+1},\ \tau_j \tau_{j+1}^\dagger$ |

The coefficients generated by the deformation (90) are in general complex

$$f = \frac{2}{9}\left[ 2\left( rs + \omega \frac{r}{s^2} + \omega^* \frac{s}{r^2} \right) - \left( \frac{1}{rs} + \omega^* \frac{r^2}{s} + \omega \frac{s^2}{r} \right) \right], \tag{92}$$

$$g_1 = -\frac{2}{9}\left[ \omega\left( \frac{r^2}{s} + \frac{s}{r^2} \right) + \omega^*\left( \frac{s^2}{r} + \frac{r}{s^2} \right) + rs + \frac{1}{rs} \right], \tag{93}$$

$$g_2 = \frac{1}{9}\left[ 3 + \left( \frac{1}{rs} + \frac{s^2}{r} + \frac{r^2}{s} \right) - 2\left( rs + \frac{s}{r^2} + \frac{r}{s^2} \right) \right]. \tag{94}$$

Even though there is some elegance in the generality of this model, these complex coefficients are not very practical. Therefore in the next sub-sections we discuss two specific cases.

### 6.1.1  $\mathbb{Z}_3$-ANNNP model: real coefficients reproducing Reference [3]

The first example features purely real coefficients. This model was originally obtained by direct calculation by Mahyaeh and Ardonne [3]. We rediscover it by considering the deformation (90) with $s = r$, ie,

$$m_j = \begin{pmatrix} 1 & & \\ & r & \\ & & r \end{pmatrix}, \quad L_{j,j+1} = \sigma_j - \sigma_{j+1}, \quad C_{j,j+1} = K_j K_{j+1}, \quad k_j = \begin{pmatrix} r & & \\ & 1 & \\ & & r^{-1} \end{pmatrix}, \tag{95}$$

such that the coefficients become

$$f = \frac{2(1+2r)(1-r^3)}{9r^2}, \quad \epsilon = \frac{2(1+r+r^2)^2}{9r^2}, \tag{96}$$

$$g_1 = -\frac{2(1-r)^2(1+r+r^2)}{9r^2}, \quad g_2 = \frac{(1-r)^2(1-2r-2r^2)}{9r^2}. \tag{97}$$

The exact ground states originally constructed in Reference [3] follow by direct application of Theorem 1. Furthermore, in Appendix B.2 we prove that the model possesses an energy gap above these ground states at least in the interval $\sqrt{\frac{3}{2}\sqrt{2}-2} \approx 0.3483 \lesssim r \lesssim 3.9912$. Finally we note that for $r = (\sqrt{3}-1)/2 \approx 0.366$ the model (91) simplifies as the parameter $g_2$ vanishes.

### 6.1.2   $\mathbb{Z}_3$-ANNNP with ground state deformation of $\mathbb{Z}_3$-XY model

For the second example we consider the deformation that we encountered before for $\mathbb{Z}_3$-XY, namely $s = r^2$,

$$m_j = \begin{pmatrix} 1 & & \\ & r & \\ & & r^2 \end{pmatrix}. \tag{98}$$

Thus the deformed ground states are identical to the ones for $\theta = 0$ discussed in Section 5.3. However, due to the non-trivial choice for $C_{j,j+1}$ the Hamiltonian will differ, specifically we obtain (91) with the coefficients

$$f = \omega^* \frac{(1-r^3)\left[(1-r^3)+3\sqrt{3}i(1+r^3)\right]}{9r^3}, \quad g_1 = -2\omega g_2, \quad g_2 = -\frac{(1-r^3)^2}{9r^3}. \tag{99}$$

The coefficient $g_1$ can be chosen to be real via a gauge transformation, ie, a permutation of diagonal elements of $m_j$.

## 6.2   $\mathbb{Z}_p$-ANNNP model: most general real coefficients

For general $\mathbb{Z}_p$ we discuss the case when all coefficients take real values. From (88) and (89) we recognise that the coefficient of $\tau_j^l \tau_{j+1}^{l'}$ is $\Delta_l \Gamma_{l'} + \Gamma_l \Delta_{l'}$. This is real for example if $\Gamma_l^* = \Delta_l$, which yields the constraints (recall that $\alpha_k > 0$)

$$\alpha_{-k} \equiv \alpha_{p-k} = \alpha_k, \tag{100}$$

for all $k$. Thus there are $(p-1)/2$ real degrees of freedom for $p$ odd and $p/2$ for $p$ even. The expansion is still not in a compact form. In Section 6.3 we will discuss a Hamiltonian with a compact form for $p$ even. For $p$ odd we did not obtain a simple compact form, except for the case $p = 3$ discussed in the next section.

The condition (100) has another consequence. Under charge conjugation

$$\sigma_j \to \sigma_j^\dagger, \quad \tau_j \to \tau_j^\dagger, \tag{101}$$

we see that

$$M_j^{-1}\sigma_j^\dagger M_j \sigma_j \to K_j, \quad K_j \to M_j^{-1}\sigma_j^\dagger M_j \sigma_j. \tag{102}$$

In this particular case the Hamiltonian (87) is invariant under charge conjugation, and together with the $\mathbb{Z}_p$-symmetry generated by $\omega^P$, the full symmetry group is the dihedral group $D_p$ [71, 72]. Note that for $p = 3$ the dihedral group is isomorphic to the symmetric group $S_3$ of all permutations.

## 6.3 $\mathbb{Z}_{2q}$-ANNNP model: compact form with real coefficients

For even $p = 2q$ it is possible to construct a model depending on a single parameter which possesses real coefficients and a simple closed form. We start with

$$m_j = \mathrm{diag}(1, r, \ldots, r^{q-1}, r^q, r^{q-1}, \ldots, r), \quad k_j = \mathrm{diag}(r, \ldots, r, r^{-1}, \ldots, r^{-1}). \tag{103}$$

Using Equation (67) we see that

$$M_j^{-1} \sigma_j^\dagger M_j \sigma_j = \sum_{l=0}^{p-1} \left[ r^{-1} + (-1)^l r \right] \theta^q(l) \tau_j^l, \tag{104}$$

$$K_j = \sum_{l=0}^{p-1} (-\omega)^l \left[ r^{-1} + (-1)^l r \right] \theta^q(l) \tau_j^l, \tag{105}$$

where

$$\theta^q(l) = \frac{1}{2q} \sum_{k=1}^q \omega^{-kl} = \begin{cases} \frac{1}{2}, & \text{if } l = 0, \\ 0, & \text{if } l \text{ even} \neq 0, \\ \frac{1}{2q} \sum_{k=1}^q \omega^{-kl}, & \text{if } l \text{ odd}. \end{cases} \tag{106}$$

Note that both (104) and (105) only contribute odd powers of $\tau$ (or the identity), hence the last two terms in (87) can only give odd powers of $\tau_j$-operators. The full expression becomes

$$M_j^{-1} \sigma_j^\dagger M_j \sigma_j k_{j+1} + k_j M_{j+1}^{-1} \sigma_{j+1}^\dagger M_{j+1} \sigma_{j+1}$$
$$= \sum_{l,l'} \left[ (-\omega)^l + (-\omega)^{l'} \right] \left[ r^{-1} + (-1)^l r \right] \left[ r^{-1} + (-1)^{l'} r \right] \theta^q(l) \theta^q(l') \tau_j^l \tau_{j+1}^{l'}. \tag{107}$$

Let us consider the different terms individually. First, the term with $l = l' = 0$ turns into an energy shift given by

$$\epsilon = \frac{(r + r^{-1})^2}{2}. \tag{108}$$

Second, the terms with $l = 0$ or $l' = 0$ turn into a magnetic-field term of the form $-\frac{f}{2}(\tau_j^l + \tau_{j+1}^l)$ for odd $l$, with the prefactor given by

$$f = -(r^{-2} - r^2)(1 - \omega^l)\theta^q(l) = \frac{r^2 - r^{-2}}{2q} \sum_{k=1}^q (\omega^{-kl} - \omega^{-(k-1)l}) = \frac{r^{-2} - r^2}{q}. \tag{109}$$

Finally, the remaining terms with $l, l' \neq 0'$ yield the terms $U_{ll'} \tau_j^l \tau_{j+1}^{l'}$ with

$$U_{ll'} = -\left( \omega^l + \omega^{l'} \right) \left( r - r^{-1} \right)^2 \theta^q(l) \theta^q(l'). \tag{110}$$

Note that $\left[ \omega^l \theta^q(l) \theta^q(l') \right]^* = \omega^{l'} \theta^q(l) \theta^q(l')$ and therefore $U_{ll'}^* = U_{ll'} = U_{l'l}$, such that the full local Hamiltonian becomes

$$\tilde{H}_{j,j+1} = -\left( \sigma_j \sigma_{j+1}^\dagger + \sigma_j^\dagger \sigma_{j+1} \right) - \frac{f}{2} \sum_{\substack{l=1 \\ l \text{ odd}}}^{p-1} \left( \tau_j^l + \tau_{j+1}^l \right) + \sum_{\substack{l,l'=1 \\ l,l' \text{ odd}}}^{p-1} U_{ll'} \tau_j^l \tau_{j+1}^{l'} + \epsilon. \tag{111}$$

We note that the Hamiltonian for even $p$ is invariant under $r \to 1/r$ and $\tau \to -\tau$.

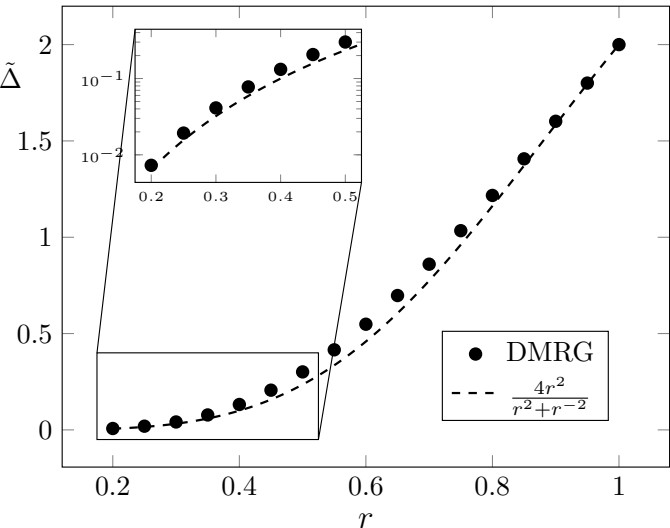

Figure 1: Energy gap $\tilde{\Delta}$ above the four-fold degenerate ground states $|\tilde{\Psi}_i\rangle$ of the frustration-free ANNNP model (112). The dots show the energy gap obtained by extrapolating the finite-size data for $L = 64, 76, 88, 100$ to the thermodynamic limit. The dashed line is the lower bound for the energy gap proven to exist in Appendix B.3. Inset: Zoom in on the small-$r$ region, logarithmic scale.

## 6.4 $\mathbb{Z}_4$-ANNNP model

The first new non-trivial example originating from the construction of the previous section is obtained for $p = 4$. In this case the local Hamiltonian becomes remarkably simple

$$\tilde{H}_{j,j+1} = -\left[\sigma_j \sigma_{j+1}^\dagger + \frac{f}{2}(\tau_j + \tau_{j+1}) - U \tau_j \tau_{j+1} + \text{h.c.}\right] + \epsilon, \tag{112}$$

with the parameters

$$f = \frac{r^{-2} - r^2}{2}, \quad U = \frac{(r - r^{-1})^2}{4}, \quad \epsilon = \frac{(r + r^{-1})^2}{2}. \tag{113}$$

Note the absence of terms like $\tau_j^2$, $\tau_j \tau_{j+1}^2$ and $\tau_j \tau_{j+1}^\dagger$, in contrast to the frustration-free $\mathbb{Z}_3$-ANNNP model (91). The correlation functions in the four-fold degenerate ground states $|\tilde{\Psi}_i\rangle$ are identical to the ones in the ANNNI model, see Equation (25),

$$\left|\frac{\langle\tilde{\Psi}_i|\sigma_j \sigma_{j'}^\dagger|\tilde{\Psi}_i\rangle}{\langle\tilde{\Psi}_i|\tilde{\Psi}_i\rangle}\right| = \frac{4}{(r + r^{-1})^2}. \tag{114}$$

In Appendix B.3 we prove that the model (112) possesses an energy gap $\tilde{\Delta}$ above the ground states. More specifically, we show that the lower bound for the gap in the thermodynamic limit is given by $\frac{4\min(r^2, r^{-2})}{r^2 + r^{-2}} \leq \tilde{\Delta}$. For completeness in Figure 1 we compare this to numerical results for the energy gap. The latter were obtained by extrapolating finite-size data from system sizes $L = 64, 76, 88, 100$ to $L \to \infty$, with the finite-size results being calculated by employing the density matrix renormalisation group (DMRG) method [39,73] using the TeNPy [74] library.

    Closer inspection of the parameters (113) shows that they satisfy the relation $f = 2\sqrt{U(1 + U)}$, which is identical to the relation along the Peschel–Emery line in the ANNNI model. This points towards a closer relation between the models (112) and (22), which we discuss in the following. In fact, even away from the frustration-free line one can map the $\mathbb{Z}_4$-ANNNP chain to two decoupled ANNNI chains. For simplicity we consider an infinitely long

system (ie, we ignore the boundary conditions) and drop the constant energy shift $\epsilon$; thus (112) turns into the Hamiltonian

$$H_{\text{ANNNP}} = -\sum_j \left( \sigma_j \sigma_{j+1}^\dagger + f\tau_j - U\tau_j\tau_{j+1} + \text{h.c.} \right). \tag{115}$$

Introducing the dual operators via

$$\sigma_j^\dagger \sigma_{j+1} \to \tilde{\tau}_j, \quad \tau_j \to \tilde{\sigma}_{j-1}\tilde{\sigma}_j^\dagger, \tag{116}$$

which satisfy the clock algebra (60) with $p = 4$, we can rewrite this as

$$H_{\text{ANNNP}}^{\text{dual}} = -\sum_j \left( \tilde{\tau}_j + f\tilde{\sigma}_j\tilde{\sigma}_{j+1}^\dagger - U\tilde{\sigma}_j\tilde{\sigma}_{j+2}^\dagger + \text{h.c.} \right). \tag{117}$$

Next we introduce two sets of Pauli matrices $\sigma_{i,j}^{x/z}$, $i = 1,2$, per lattice site $j$, and consider the mapping [72, 75, 76]

$$\tilde{\sigma}_j = e^{i\frac{\pi}{4}} \left( \frac{\sigma_{1,j}^x - i\sigma_{2,j}^x}{\sqrt{2}} \right), \quad \tilde{\tau}_j + \tilde{\tau}_j^\dagger = \sigma_{1,j}^z + \sigma_{2,j}^z. \tag{118}$$

From the second relation in (118) we can already infer that the $\tilde{\tau}_j$-terms are mapped to a transverse magnetic field on the Ising ladder. For the other terms, we use the following simple identity

$$\tilde{\sigma}_j\tilde{\sigma}_{j+j'}^\dagger + \text{h.c.} = \sigma_{1,j}^x\sigma_{1,j+j'}^x + \sigma_{2,j}^x\sigma_{2,j+j'}^x. \tag{119}$$

Thus the dual of the $\mathbb{Z}_4$-ANNNP model can be written as the sum of two decoupled ANNNI chains

$$H_{\text{ANNNP}}^{\text{dual}} = H_{\text{ANNNI},1}^{\text{dual}} + H_{\text{ANNNI},2}^{\text{dual}}, \tag{120}$$

with

$$H_{\text{ANNNI},i}^{\text{dual}} = -\sum_j \left( \sigma_{i,j}^z + f\sigma_{i,j}^x\sigma_{i,j+1}^x - U\sigma_{i,j}^x\sigma_{i,j+2}^x \right). \tag{121}$$

Performing another duality transformation (121) can be brought into the form (22) discussed in Section 3.2. The condition for the parameters $f$ and $U$ to be on the frustration-free line directly turns into the Peschel–Emery line for the two ANNNI models.

## 6.5  $\mathbb{Z}_6$-ANNNP model

Interestingly, in the case $p = 6$ the deformation (103) leads to another rather simple model with the local Hamiltonian

$$\tilde{H}_{j,j+1} = -\left[ \sigma_j\sigma_{j+1}^\dagger + \frac{f}{2}\left( \tau_j + \frac{1}{2}\tau_j^3 + \tau_{j+1} + \frac{1}{2}\tau_{j+1}^3 \right) \right.$$
$$\left. -U\left( \tau_j\tau_{j+1} + \frac{1}{4}\tau_j\tau_{j+1}^3 + \frac{1}{4}\tau_j^3\tau_{j+1} - \frac{1}{2}\tau_j\tau_{j+1}^\dagger + \frac{1}{8}\tau_j^3\tau_{j+1}^3 \right) + \text{h.c.} \right] + \epsilon, \tag{122}$$

where the parameters are given by

$$f = \frac{r^{-2} - r^2}{3}, \quad U = \frac{2}{9}(r - r^{-1})^2, \quad \epsilon = \frac{(r + r^{-1})^2}{2}. \tag{123}$$

We note that even though the $\mathbb{Z}_6$-symmetry allows a wealth of terms of the form $\tau_j^l\tau_{j+1}^{l'}$, along the frustration-free line the relative prefactors of them are fixed to fairly simple values. In

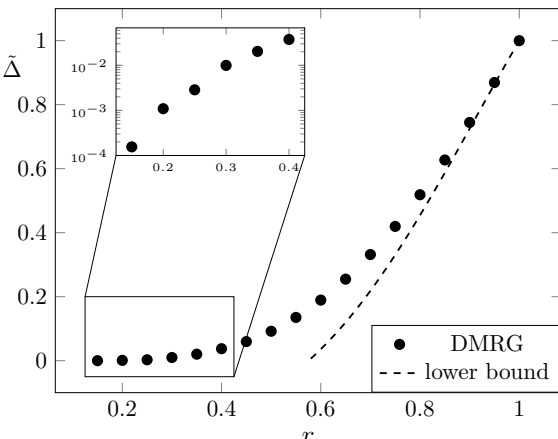

Figure 2: Energy gap $\tilde{\Delta}$ above the six-fold degenerate ground states of the frustration-free ANNNP model (122). The dots show the energy gap obtained by extrapolating the finite-size data for $L = 64, 76, 88, 100$ to the thermodynamic limit. The dashed line is the lower bound for the energy gap obtained in Appendix B.4.

Figure 2 we show the energy gap above the six-fold degenerate ground state. The numerical results were obtained by extrapolation from finite-size data, they clearly indicate the existence of a finite energy gap along the frustration-free line. In addition, in Appendix B.4 we prove that the model is gapped at least in the interval $0.5754 \lesssim r \lesssim 1/0.5754 \approx 1.7379$. We note in passing that using more advanced methods for open boundary conditions [52] it is possible to enlarge the region for which the existence of a finite energy gap can be proven. However, the obtained lower bounds are found to be quite small ($< 10^{-5}$).

# 7 Discussion

We have presented a constructive approach to understand and derive one-dimensional frustration-free spin models. Starting from a simple point, for example a classical system, we derived the corresponding frustration-free quantum models and their exact ground states. We have shown that many known frustration-free spin-1/2, spin-1 and $\mathbb{Z}_p$-clock models can be understood in this framework on an equal footing. Hence our approach provides an overarching framework for many frustration-free systems.

More specifically, the approach allowed us to connect two distinct frustration-free $\mathbb{Z}_3$-clock models recently introduced by Iemini et al. [2] and Mahyaeh and Ardonne [3]. As we have shown, both models can be interpreted as different deformations of the classical three-state Potts chain, see Figure 3 for an illustration of their relation. As a side remark, we analytically showed that the energy gap remains finite in a finite region around the classical point for both models. This in particular implies that both models (or their parafermion analogues) are in the same (topological) phase. Furthermore, we have constructed several new frustration-free $\mathbb{Z}_p$-clock models, including $\mathbb{Z}_4$- and $\mathbb{Z}_6$-generalisations of the Peschel–Emery line of the original ANNNI chain.

We stress that the list of frustration-free clock models considered above is by no means extensive. On the contrary, the examples discussed here should be regarded as a proof of principle on how to apply the general construction. Several generalisations come to mind: First, one may consider chiral classical models [77, 78] as starting points in the deformation construction. However, since in this case the local Hamiltonians are no longer given by simple projectors, the deformed Hamiltonians so obtained may become quite complicated. Second,

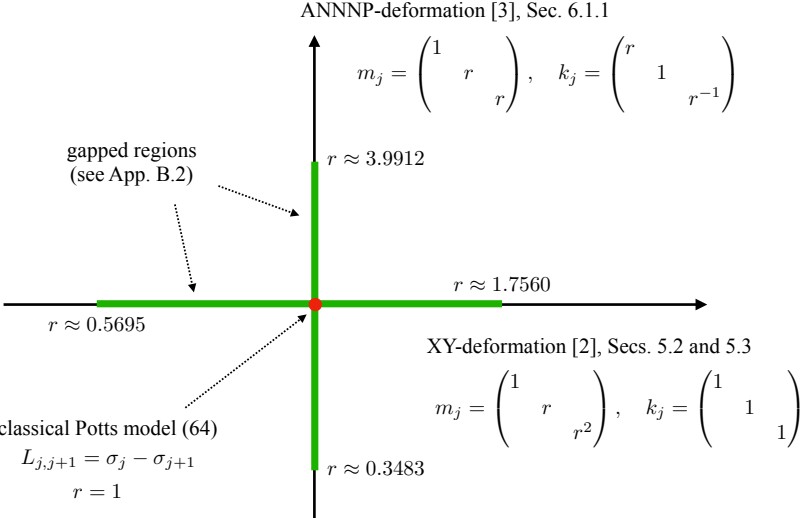

Figure 3: Schematic sketch of the relation between the two frustration-free $\mathbb{Z}_3$-clock models introduced by Iemini et al. [2] (see Sections 5.2 and 5.3) and Mahyaeh and Ardonne [3] (see Section 6.1.1). Both models can be obtained as deformations of the classical three-state Potts chain (red dot) using the local deformations $m_j$ and central term $k_j$ depending on the parameter $r$. The green lines indicate the regions in which the systems are proven to be gapped in Appendix B.2. In particular, within this region the two models can be connected without closing the energy gap, implying that they are in the same phase.

in this paper we have kept the considered deformations to be homogeneous, a restriction that is not required by Theorem 1. Thus our results can be extended to inhomogeneous systems. Third, another generalisation would be to relax the requirement for the operator $C_{j,j+1}$ to be positive definite. In such a case, the ground states of the undeformed model are no longer transformed into ground states of the new model. However, they will still be exact eigenstates, potentially in the middle of the spectrum, and thus may be relevant in the context of quantum many-body scars [79–84].

# Acknowledgements

We thank Floris Elzinga, Vladimir Korepin, Marius Lemm and Iman Mahyaeh for useful discussions and correspondence. H. K. was supported in part by JSPS Grant-in-Aid for Scientific Research on Innovative Areas No. JP18H04478 and JP20H04630, and JSPS KAKENHI Grant No. JP18K03445. This work is part of the D-ITP consortium, a program of the Netherlands Organisation for Scientific Research (NWO) that is funded by the Dutch Ministry of Education, Culture and Science (OCW).

# A  Witten's conjugation argument

In this appendix we recall Witten's original conjugation argument on the ground-state degeneracy of supersymmetric Hamiltonians. Consider two supercharges $Q$ and $Q^\dagger$ as well as a

Hamiltonian $H$ satisfying

$$Q^2 = (Q^\dagger)^2 = 0, \quad H = Q^\dagger Q + Q Q^\dagger. \tag{124}$$

First we note that any zero-energy ground state $|\psi\rangle$ of $H$ is annihilated by both $Q$ and $Q^\dagger$. Furthermore, it is not possible to obtain $|\psi\rangle$ by action of $Q$, ie, $|\psi\rangle \neq Q|\phi\rangle$ for any state $|\phi\rangle$. (To see this assume $|\psi\rangle = Q|\phi\rangle$. But since $|\psi\rangle$ is a zero-energy ground state we have $0 = Q^\dagger|\psi\rangle = Q^\dagger Q|\phi\rangle$ which implies $\langle\phi|Q^\dagger Q|\phi\rangle = \|Q|\phi\rangle\|^2 = 0$ and thus $Q|\phi\rangle = 0$ in contradiction with the assumption that $|\psi\rangle$ is a ground state.)

Now let us consider the deformed/conjugated operators $\tilde{Q} = MQM^{-1}$, $\tilde{Q}^\dagger = (\tilde{Q})^\dagger$ and $\tilde{H} = \tilde{Q}^\dagger\tilde{Q} + \tilde{Q}\tilde{Q}^\dagger$ with $M$ being invertible. Obviously, if $|\psi\rangle$ is a ground state of $H$, the deformed state $|\tilde{\psi}\rangle = M|\psi\rangle$ is annihilated by $\tilde{Q}$. Furthermore, $|\tilde{\psi}\rangle$ cannot be written as $|\tilde{\psi}\rangle = \tilde{Q}|\tilde{\phi}\rangle$ for any $|\tilde{\phi}\rangle$, since this would imply that $|\psi\rangle = QM^{-1}|\tilde{\phi}\rangle$ in contradiction with the assumption that $|\psi\rangle$ was a ground state of $H$. Thus $|\tilde{\psi}\rangle$ is a ground state of $\tilde{H}$ establishing a one-to-one correspondence between the ground-state manifolds of $H$ and $\tilde{H}$.

# B   Energy gap of some $\mathbb{Z}_3$-, $\mathbb{Z}_4$- and $\mathbb{Z}_6$-models

The conjugation argument does only provide information about the ground-state manifold. In order to obtain information about the energy gap above it, additional techniques have to be employed. In Appendix B.1 we recall Knabe's method [54], which was originally applied to the AKLT model with periodic boundary conditions. This is then applied in Appendices B.2, B.3 and B.4 to prove the existence of an energy gap in specific $\mathbb{Z}_3$-, $\mathbb{Z}_4$-, and $\mathbb{Z}_6$-models.

## B.1   Knabe's method

We consider a system with $N$ sites, open boundary conditions and the Hamiltonian

$$H_N = \sum_{j=1}^{N-1} P_{j,j+1}, \tag{125}$$

with the $P_{j,j+1}$ being two-site projection operators. We assume $\bigcap_j \ker(P_{j,j+1}) \neq \{0\}$, ie, the ground state is at zero energy, and denote the energy gap of $H_N$ by $\Delta_N$. Then we have

**Theorem 2** (Knabe's method [54]). *For the projector Hamiltonian $H_N$ the gap above the ground state ($\Delta_N$) is bounded from below by*

$$\Delta_N \geq \frac{m-1}{m-2}\left(\min_{m'=2,\dots,m}\{\Delta_{m'}\} - \frac{1}{m-1}\right), \tag{126}$$

*where $\Delta_{m'}$ denotes the gap of the $m'$-site, sub-system Hamiltonian*

$$h_{j,m'} = \sum_{k=j}^{j+m'-2} P_{k,k+1}. \tag{127}$$

*Proof.* Note that $H_N$ is positive semi-definite, therefore $H_N^2 \geq \Delta_N H_N$. In other words, if we obtain the above inequality with $\Delta_N$, the Theorem is proven. We have the analogous statement for $h_{j,m'}$, $h_{j,m'}^2 \geq \Delta_m h_{j,m'}$, and moreover realise that $P_{j,j+1}^2 = P_{j,j+1}$ and $[P_{j,j+1}, P_{k,k+1}] = 0$ for $|j-k| > 1$.

To prove the bound, we first expand $H_N^2$

$$H_N^2 = \sum_{j=1}^{N-1} h_{j,2}^2 + \sum_{m'=1}^{m-2} \sum_{j=1}^{N-m'-1} \left(h_{j,2} h_{j+m',2} + \text{h.c.}\right) + \sum_{|j-k|>m-2} h_{j,2} h_{k,2} \tag{128}$$

$$\geq H_N + \sum_{m'=1}^{m-2} \frac{m-m'-1}{m-2} \sum_{j=1}^{N-m'-1} \left(h_{j,2} h_{j+m',2} + \text{h.c.}\right), \tag{129}$$

with the second step following from the fact that $h_{j,2} h_{k,2}$ is positive semi-definite for $|k-j| > 1$. This can be further reduced to

$$H_N^2 \geq H_N + \frac{1}{m-2} \left[ \sum_{j=1}^{N-m+1} h_{j,m}^2 + \sum_{m'=2}^{m-1} \left(h_{1,m'}^2 + h_{N-m'+1,m'}^2\right) - (m-1)H_N \right] \tag{130}$$

$$\geq \left(1 - \frac{m-1}{m-2}\right) H_N + \frac{1}{m-2} \left[ \Delta_m \sum_{j=1}^{N-m+1} h_{j,m} + \sum_{m'=2}^{m-1} \Delta_{m'} \left(h_{1,m'} + h_{N-m'+1,m'}\right) \right]. \tag{131}$$

Because we have the expansion

$$H_N = \frac{1}{m-1} \left[ \sum_{j=1}^{N-m+1} h_{j,m} + \sum_{m'=2}^{m-1} \left(h_{1,m'} + h_{N-m'+1,m'}\right) \right] \tag{132}$$

and $\Delta_{m'} \leq 1$, the last term of (131) can be simplified to obtain

$$H_N^2 \geq -\frac{1}{m-2} H_N + \frac{m-1}{m-2} \min_{m'=2,\ldots,m} \{\Delta_{m'}\} H_N \tag{133}$$

$$= \frac{m-1}{m-2} \left( \min_{m'=2,\ldots,m} \{\Delta_{m'}\} - \frac{1}{m-1} \right) H_N. \tag{134}$$

This proves the Theorem with the lower bound $\Delta_N \geq \frac{m-1}{m-2} \left( \min_{m'=2,\ldots,m} \{\Delta_{m'}\} - \frac{1}{m-1} \right)$. $\qquad\square$

**Remark 3.** *Given that the models considered here can be viewed as parent Hamiltonians for matrix product ground states, one can apply more powerful tools [33, 50, 52] to prove the existence of energy gaps.[6] These methods allow one to extend the parameter regions with proven energy gaps. However, in some cases, Knabe's method gives us a better lower bound for an energy gap for fixed parameters. We also note that when analysing the energy gap, special care has to be taken regarding the treatment of different boundary conditions.*

Note that the argument above assumes the Hamiltonian to be the sum of projectors. The systems studied in our paper do not fit that picture. However, since they are frustration-free we can still obtain a bound using the following observation:

**Corollary 4.** *For a frustration-free model with an n-fold degenerate zero-energy ground state and a p-dimensional local Hilbert space, with $p^2 > n$, we can arrange the two-site eigenvalues $\tilde{\Delta}_2^k$ and*

---

[6]For example, for a so-called injective matrix product state one can prove that the corresponding parent Hamiltonian has a unique ground state and a finite energy gap [33, 34, 50]. However, most of the ground states we have looked at in this article do not qualify as injective (see, eg, Reference [16] for the ANNNI model), because of the degeneracy.

*normalised eigenstates $|\tilde{\psi}_k\rangle$ such that $\tilde{\Delta}_2^k \leq \tilde{\Delta}_2^l$ for $k < l$ and $\tilde{\Delta}_2^1, \ldots, \tilde{\Delta}_2^n = 0$. Then two-site Hamiltonian can be bounded from below as follows,*

$$\tilde{H}_{j,j+1} = \sum_{k=n+1}^{p^2} \tilde{\Delta}_2^k |\tilde{\psi}_k\rangle \langle \tilde{\psi}_k| = \tilde{\Delta}_2^{n+1} \sum_{k=n+1}^{p^2} |\tilde{\psi}_k\rangle \langle \tilde{\psi}_k| + \sum_{k=n+1}^{p^2} \left( \tilde{\Delta}_2^k - \tilde{\Delta}_2^{n+1} \right) |\tilde{\psi}_k\rangle \langle \tilde{\psi}_k| \quad (135)$$

$$\geq \tilde{\Delta}_2^{n+1} P_{j,j+1} = \tilde{\Delta}_2 P_{j,j+1}, \quad (136)$$

*with the gap $\tilde{\Delta}_2 = \tilde{\Delta}_2^{n+1}$ of the frustration-free Hamiltonian $\tilde{H}_{j,j+1}$ and $P_{j,j+1}$ denoting the projector onto the space orthogonal to its ground-state manifold. The min-max theorem [85] then implies for the gap $\tilde{\Delta}_N$ of the frustration-free model on $N$ sites*

$$\tilde{\Delta}_N \geq \tilde{\Delta}_2 \Delta_N. \quad (137)$$

Thus in order to prove that a frustration-free Hamiltonian possesses an energy gap $\tilde{\Delta}$ above its ground states in the thermodynamic limit, we proceed as follows: (i) We consider projectors $P_{j,j+1}$ onto the space orthogonal to the local ground states on the lattice sites $j$ and $j+1$ and determine the gap $\Delta_2$ above these ground states. (ii) From that we construct the auxiliary $m$-site Hamiltonian $h_{1,m} = \sum_{j=1}^m P_{j,j+1}$ and determine its energy gap $\Delta_m$. (iii) If this gap satisfies $\min_{m'=2,\ldots,m} \{\Delta_{m'}\} > 1/(m-1)$, then the auxiliary $N$-site Hamiltonian $H_N$ will have a gap $\Delta_N$ satisfying (126). (iv) Due to (137) the gap $\tilde{\Delta}$ of the original frustration-free Hamiltonian is bounded from below by

$$\tilde{\Delta} = \lim_{N \to \infty} \tilde{\Delta}_N \geq \lim_{N \to \infty} \tilde{\Delta}_2 \Delta_N \geq \tilde{\Delta}_2 \frac{m-1}{m-2} \left( \min_{m'=2,\ldots,m} \{\Delta_{m'}\} - \frac{1}{m-1} \right). \quad (138)$$

Every $m > 2$ gives a lower bound on the gap, so the supremum over subsysten sizes is also a lower bound. Usually, the bound increases for increasing $m$. Since the computation of $\Delta_m$ requires exact diagonalization of a $p^m \times p^m$ matrix, the maximal feasible $m$ is constrained by computational resources. In the following appendices we apply this line of argument to several models.

## B.2 Gap in $\mathbb{Z}_3$-models

In order to treat both $\mathbb{Z}_3$-models (84) (for $\theta = 0$) and the models discussed in Section 6.1 within the same framework, we consider the general, diagonal deformation with

$$m_j = \begin{pmatrix} 1 & & \\ & r & \\ & & s \end{pmatrix}, \quad (139)$$

where $r, s > 0$. For each point in the $(r,s)$-plane we get a lower bound on the thermodynamic gap by means of (126), provided that for some feasible $m$ the relevant energy gap of the auxiliary $m$-site Hamiltonian satisfies $\min_{m'=2,\ldots,m} \{\Delta_{m'}\} > 1/(m-1)$. Computational resources allow us to go up to $m = 7$. In Figure 4 we have depicted the maximal lower bound for $m = 3, \ldots, 7$ in the $(r,s)$-plane obtained from this. Note that this is a lower bound for the gap of the auxiliary projector Hamiltonian. For a particular parent Hamiltonian like (84) and (91), the true gap depends on the local gap $\tilde{\Delta}_2$. As long as the local parent Hamiltonian has the same degeneracy as the local auxiliary Hamiltonian it is gapped for the same parameter regime, by virtue of (138). We only consider the triangle $s \leq r \leq 1$, since due to the dihedral symmetry of the model there is a six-fold symmetry in the $(r,s)$-plane. The red line denotes the boundary of the region that is definitely gapped, ie, for all points above this line in the $(r,s)$-plane it is assured that the full system is gapped in the thermodynamic limit. The blue

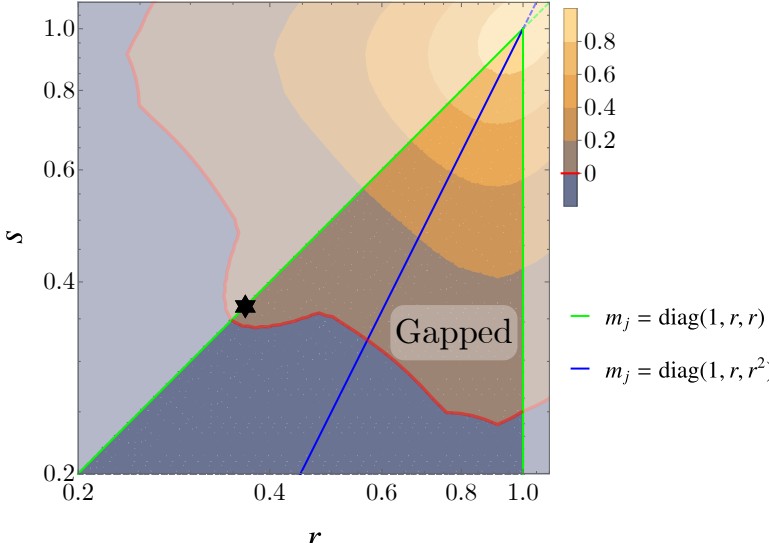

Figure 4: Log-log contour plot of the lower bound $\max_{m=3,\dots,7} \frac{m-1}{m-2}\left(\min_{m'=2,\dots,m}\{\Delta_{m'}\} - \frac{1}{m-1}\right)$ for the deformation in Equation (139). For a finite lower bound the system is gapped in the thermodynamic limit $N \to \infty$, ie, all points above the red line yield gapped systems. The blue and green lines correspond to the ground states of (84) (for $\theta = 0$) and (91), respectively. The star is the special point with $g_2 = 0$.

and green lines correspond to the ground states of (84) (for $\theta = 0$) and (91), respectively, with the black star indicating the model (91) at the special point $g_2 = 0$. Given the six-fold symmetry in the $(r,s)$-plane, we have to be careful how to display the green $(1,r,r)$ and blue $(1,r,r^2)$ lines. For the blue line, note that $(1,r,r^2) \simeq (r^{-2}, r^{-1}, 1)$, since the Hamiltonian is invariant under rescaling of $M$. Also the freedom in the form of the dihedral symmetry lets us write $(1,r,r^2) \simeq (1, r^{-1}, r^{-2})$, permuting the entries. Hence the blue line for $r > 1$, maps to the blue line for $r < 1$ under the symmetry. Using the same reasoning for the green line we obtain $(1,r,r) \simeq (r^{-1}, 1, 1) \simeq (1, 1, r^{-1})$, mapping $(1,r,r)$ for $r > 1$ to $(1,1,s)$ for $s = r^{-1}$.

Let us zoom in on the two lines $s = r^2$ and $s = r$ that correspond to the ground states of (84) and (91) respectively. In Table 1 we list the lower and upper limit $r^{\text{low,up}}$ for the gapped region for different sub-system sizes $m$. For $s = r^2$ the upper limit is simply $r^{\text{up}} = 1/r^{\text{low}}$, as follows from the symmetry discussed above. As $m$ increases we see that the region increases in both directions.

On the other hand, for $s = r$ something peculiar occurs. The lower limit $r^{\text{low}}$ is significantly better for $m = 3$ than for $m = 4,\dots,7$. This lower limit has the exact value of $r^{\text{low}} = 2^{1/4} - 2^{-1/4} = \sqrt{\frac{3}{2}\sqrt{2} - 2} \approx 0.3483$ The upper limit, on the other hand, does become more informative as $m$ increases.

In total we deduce that the full system (84) (for $\theta = 0$) is gapped in the thermodynamic limit for $0.5695 \lesssim r \lesssim 1/0.5695$ and (91) for $0.3483 \lesssim r \lesssim 3.9912$. In particular this implies that in this parameter regime the models can be adiabatically connected to the classical model obtained for $r = s = 1$ as sketched in Figure 3.

Table 1: Lower and upper limit $r^{\text{low,up}}$ for the gapped regions of the $\mathbb{Z}_3$-XY model (84) and $\mathbb{Z}_3$-ANNNP model (91) as deduced from different sub-system sizes $m$. The bold values indicate the extremal values which are stated in the main text.

| $m$ | $s = r^2$ ($\mathbb{Z}_3$-XY model) | | $s = r$ ($\mathbb{Z}_3$-ANNNP model) | |
|---|---|---|---|---|
| | $r^{\text{low}}$ | $r^{\text{up}}$ | $r^{\text{low}}$ | $r^{\text{up}}$ |
| 3 | 0.6337 | 1.5779 | **0.3483** | 2 |
| 4 | 0.6204 | 1.6119 | 0.4216 | 2.6796 |
| 5 | 0.6026 | 1.6595 | 0.4259 | 3.0146 |
| 6 | 0.5853 | 1.7086 | 0.4200 | 3.6233 |
| 7 | **0.5695** | **1.7560** | 0.4116 | **3.9912** |

### B.3   Gap in $\mathbb{Z}_4$-ANNNP model

We can apply the same method to analyse the gap of the $\mathbb{Z}_4$-ANNNP model (112). For this model it is sufficient to consider $m = 3$, since

$$\Delta_3 = \frac{1}{2} + \frac{\min(r^2, r^{-2})}{r^2 + r^{-2}}, \tag{140}$$

which is strictly larger than $1/2$ for $0 < r < \infty$. Thus we deduce for the gap in the thermodynamic limit

$$\tilde{\Delta} \geq \frac{4 \min(r^2, r^{-2})}{r^2 + r^{-2}}. \tag{141}$$

Instead of using Corollary 4, the lower bound (141) can also be obtained from the mapping to two decoupled ANNNI chains, together with the lower bound for the energy gap along the Peschel–Emery line of the ANNNI chain obtained in Reference [15].

### B.4   Gap in $\mathbb{Z}_6$-ANNNP model

For the $\mathbb{Z}_6$-ANNNP model (122) the condition of $\Delta_3 > 1/2$ shows that the model is gapped at least in the interval $0.5754 \lesssim r \lesssim 1/0.5754$, where we have used the invariance of the model under $r \to 1/r$. The region does not improve for $m = 4, 5$ and higher sub-systems sizes are not accessible with our current resources.

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
