# Peer review of "Interrelations among frustration-free models via Witten's conjugation"

_SciPost Physics Core, doi:SciPost Phys. Core 4, 027 (2021)_

## Round 1 · Referee Report · Anonymous (Referee 1) · 2021-5-25

Report

The referees have addressed all my concerns. I recommend publication of this work: it is well-written and unifies a variety of phenomena, and I believe it will guide researchers in the construction of frustration-free models.

---

## Round 1 · Referee Report · Anonymous (Referee 2) · 2021-7-7

Report

Since this is presumably the last round of reviews, I will try to focus on the big picture.

Overall, I think the value of the manuscript stems primarily from the comprehensive treatment of a large class of seemingly different models on a unified footing, as some kind of "encyclopedic" paper (Even though it is not encyclopedic in the sense that there is likely more models of this kind, but this is not a point of criticism.), while each of the individual treatments itself is quite elementary and "low-hanging fruit", and could well be posed as an exercise problem in a master-level course.

I think this it is absolutely fine and doesn't speak against the paper in any way. There is nothing wrong with harvesting low-hanging fruit in a comprehensive fashion. However, I feel it is not done in a comprehensive fashion. Half of the fruit is still on the tree, and it is not that it can only be reached by climbing high up in the tree. Still, this would be ok if it were crystal clear from reading the manuscript that there is still half the fruit on the tree, and it can be reached by simply stretching one's arm. However, my feeling is that the manuscript tells other people looking for fruit "Yes, there is still some apples on the tree, but they are super high up, so you must really have no fear of height if you want to reach them."

As I said previously, this perpetuates the legend that these type of 1D systems are hard to deal with, and I think this is not ok: People in the field should be made aware of the best tools - in particular in an encyclopedic article - since then they can apply them, and the ease or hardness of applying the tools should not be obfuscated.

To list two very clear examples where I think the manuscript falls short in that regard (unless I missed a very clear edit in that direction): 1) When talking about the correlation function, there is no mention whatsoever that using MPS & transfer matrices, one can obtain the exact form of the correlation function, and there is an exponential decay. This is a straightforward procedure, in particular if done numerically. E.g., adding a plot of the correlation length would have been minimal effort. 2) When talking about gaps, the authors state regimes where they can bound the gap (and they give bounds). However, there is no mention of the fact that the parent Hamiltonian of any injective MPS has a unique ground state + gap. Injectivity is a property which is easy to check (even analytically, e.g. by computing a determinant), generic, and will - for a smooth interpolation, as the ones considered - only break down at singular points (unless the whole path is non-injective, i.e. it has a degenerate ground state). Even more, even for non-injective cases, there is always a gap above the ground space. (Intuitively, this is linked to the fact that correlations always decay exponentially.) This is one of the key results of Refs. [32] (FNW) and [49] (Nachtergaele). (There is Remark 3 in Appendix B.1, but it gives precisely the idea that this is "more abstract", while in fact checking injectivity is less work than computing the Knabe bound.)

I think it is crucial that these things are stated clearly, as well as that they are clear from elementary properties of the state, rather than hiding them behind stating that (paraphrased) "similar things can be done using MPS", and later making it sound like all this gives is that it is much more technical and gives worse bounds on the gap (as in Sec. 6.5).

Overall, I think it is an editorial decision whether the the "encyclopedic" character of the overview of models compensates sufficiently for the fact that there is still many low-hanging apples left on the tree.

However, I feel somewhat strongly that this fact (that there are more apples, and they are easy to reach) should be made more explicit. I certainly dislike the idea of being told in the future by people that they did not apply elementary MPS techniques to problem X because applying MPS to X is hard, as can be seen by the fact that the present paper did not do it, even though they were aware of the applicability of MPS techniques.

---

## Round 1 · Author Response

Reply to referee 1

We thank the referee for his/her positive report and useful remarks. Indeed we fully agree with the referee that one of our main contributions is to elucidate the Witten conjugation method to the condensed matter community. We then use it to provide a unified framework to discuss several recently studied frustration-free models, a motivation that is now also highlighted by the revised title. We believe that the revised manuscript is clear that we do not introduce a fundamentally new method, but rather apply Witten’s conjugation to achieve this aim. In order to clarify the relation to existing MPS works we have extended the introduction (new 6th paragraph).

Reply to referee 2

We thank the referee for his/her report. We have extended the discussion on the application of Witten conjugation in the MPS framework and added further references. We have also included a discussion of alternative methods (to Knabe’s one) to proof the existence of an energy gap, and clarified Remark 3 in App. B accordingly.

We would like to reiterate that our main aim is to provide a unified framework to discuss several frustration-free models recently discussed in condensed matter physics. We achieve this by applying Witten’s conjugation arguments, which we in this way also elucidate to the condensed matter community. We believe that while this method is not new (eg, known in the MPS community), it is still fairly unknown in condensed matter physics. (This is also why we decided to keep the presentation in a notation closer to the condensed matter community.) In particular, we use this framework to clarify the interrelations between different models, as is now also highlighted by the revised title of our manuscript.

Regarding the correlation function (70) we would like to note that this has been included merely to show the general strategy to obtain such results, not because the distant-independent result (70) in itself is of physical interest. There are of course various local and non-local correlation functions one can consider, and which ones are relevant ultimately depends on the underlying setup at hand, eg, two-point spin correlation functions, Green functions of (para)fermions, etc. These correlation functions can be obtained by the reader by adapting the steps leading to (70).

---

## Round 1 · List of Changes

-revised the title
-changed “extend Witten’s conjugation” to “apply Witten’s conjugation” in the abstract
-revised 3rd paragraph of the introduction and added Refs. 35-37, 41
-slightly revised the wording in the 4th paragraph of the introduction
-added a new paragraph (new 6th) in the introduction to discuss Witten conjugation in the context of MPS states and alternative (more sophisticated) methods to prove the existence of an energy gap
-revised the wording in the paragraph after the proof of Theorem 1
-extended the discussion on the energy gap in the last paragraph of Sec. 5.2
-extended Remark 3 in App. B.1
-corrected some typos

---

## Round 2 · Author Response

We thank the referee for his/her positive report.
———————————
Reply to referee 2
We thank the referee for his/her report, in particular for acknowledging the usefulness of our manuscript in providing a unified framework to treat seemingly different models. With this in mind, we view the simplicity of the treatment of the individual systems as an advantage. Also, our aim was not to provide an encyclopedic paper picking all low-hanging fruit, as noted in the last paragraph of Section 7.
We have revised our manuscript to address the two points explicitly raised: (1) We now mention in the introduction that MPS methods yield correlation functions, and (2) we revised Remark 3 to avoid the impression that MPS are hard to deal with. We also added a footnote to link our findings to the general framework of injective MPS.

---

## Round 2 · List of Changes

Summary of changes:
-penultimate paragraph of the introduction: added a remark that correlation functions can be obtained via MPS methods
-rephrased Remark 3 in line 594
-added footnote 6 on page 24 to discuss properties of injective MPS

---

## Editorial Decision

published